# Breast Milk Mesenchymal Stem Cells and/or Derived Exosomes Mitigated Adenine-Induced Nephropathy via Modulating Renal Autophagy and Fibrotic Signaling Pathways and Their Epigenetic Regulations

**DOI:** 10.3390/pharmaceutics15082149

**Published:** 2023-08-16

**Authors:** Tarek Khamis, Amira Ebrahim Alsemeh, Asma Alanazi, Asmaa Monir Eltaweel, Heba M. Abdel-Ghany, Doaa M. Hendawy, Adel Abdelkhalek, Mahmoud A. Said, Heba H. Awad, Basma Hamed Ibrahim, Dina Mohamed Mekawy, Corina Pascu, Crista Florin, Ahmed Hamed Arisha

**Affiliations:** 1Department of Pharmacology and Laboratory of Biotechnology, Faculty of Veterinary Medicine, Zagazig University, Zagazig 44519, Egypt; t.khamis@vet.zu.edu.eg; 2Human Anatomy and Embryology Department, Faculty of Medicine, Zagazig University, Zagazig 44519, Egypt; 3College of Medicine, King Saud bin Abdulaziz University for Health Sciences (KSAU-HS), Riyadh 11481, Saudi Arabia; 4King Abdullah International Medical Research Center, Riyadh 11481, Saudi Arabia; 5Department of Pathology, Faculty of Veterinary Medicine, Zagazig University, Zagazig 44519, Egypt; 6Biochemistry and Molecular Biology Department, Faculty of Medicine, Zagazig University, Zagazig 44519, Egypt; 7Department of Food Hygiene, Safety and Technology, Faculty of Veterinary Medicine, Badr University in Cairo, Badr City 11829, Egypt; 8Zagazig University Hospital, Zagazig University, Zagazig 44511, Egypt; 9Department of Pharmacology and Toxicology, Faculty of Pharmacy, October University for Modern Sciences and Arts (MSA), Giza 12451, Egypt; 10Pathology Department, Faculty of Medicine, Zagazig University, Zagazig 44519, Egypt; 11Medical Biochemistry and Molecular Biology Department, Faculty of Medicine, Cairo University, Cairo 11562, Egypt; dinaa_dr@cu.edu.eg; 12Faculty of Veterinary Medicine, University of Life Sciences, King Mihai I from Timisoara [ULST], Aradului St. 119, 300645 Timisoara, Romania; corinapascu@usvt.ro; 13Department of Soil Science, Faculty of Agriculture, University of Life Sciences, King Mihai I from Timisoara [ULST], Aradului St. 119, 300645 Timisoara, Romania; 14Department of Animal Physiology and Biochemistry, Faculty of Veterinary Medicine, Badr University in Cairo, Badr City 11829, Egypt; 15Department of Physiology and Laboratory of Biotechnology, Faculty of Veterinary Medicine, Zagazig University, Zagazig 44511, Egypt

**Keywords:** mesenchymal stem cells, exosomes, miRNA, mRNA, long-non-coding RNA

## Abstract

Chronic kidney disease (CKD), a global health concern, is highly prevalent among adults. Presently, there are limited therapeutic options to restore kidney function. This study aimed to investigate the therapeutic potential of breast milk mesenchymal stem cells (Br-MSCs) and their derived exosomes in CKD. Eighty adult male Sprague Dawley rats were randomly assigned to one of six groups, including control, nephropathy, nephropathy + conditioned media (CM), nephropathy + Br-MSCs, nephropathy + Br-MSCs derived exosomes (Br-MSCs-EXOs), and nephropathy + Br-MSCs + Br-MSCs-EXOs. Before administration, Br-MSCs and Br-MSCs-EXOs were isolated, identified, and labeled with PKH-26. SOX2, Nanog, and OCT3/4 expression levels in Br-MSCs and miR-29b, miR-181, and Let-7b in both Br-MSCs and Br-MSCs-EXOs were assayed. Twelve weeks after transplantation, renal function tests, oxidative stress, expression of the long non-coding RNA SNHG-7, autophagy, fibrosis, and expression of profibrotic miR-34a and antifibrotic miR-29b, miR-181, and Let-7b were measured in renal tissues. Immunohistochemical analysis for renal Beclin-1, LC3-II, and P62, Masson trichome staining, and histopathological examination of kidney tissues were also performed. The results showed that Br-MSCs expressed SOX2, Nanog, and OCT3/4, while both Br-MSCs and Br-MSCs-EXOs expressed antifibrotic miR-181, miR-29b, and Let-7b, with higher expression levels in exosomes than in Br-MSCs. Interestingly, the administration of Br-MSCs + EXOs, EXOs, and Br-MSCs improved renal function tests, reduced renal oxidative stress, upregulated the renal expression of SNHG-7, AMPK, ULK-1, Beclin-1, LC3, miR-29b, miR-181, Let-7b, and Smad-7, downregulated the renal expression of miR-34a, AKT, mTOR, P62, TGF-β, Smad-3, and Coli-1, and ameliorated renal pathology. Thus, Br-MSCs and/or their derived exosomes appear to reduce adenine-induced renal damage by secreting antifibrotic microRNAs and potentiate renal autophagy by modulating SNHG-7 expression.

## 1. Introduction

Kidney disease is a significant public health concern worldwide, affecting millions of people of all ages. Chronic kidney disease (CKD) and acute kidney injury (AKI) are two distinct forms of kidney disease that significantly impact an individual’s health and well-being. CKD is an irreversible progressive loss of kidney function caused by several underlying conditions, including diabetes and hypertension. It has a higher prevalence, as, among ten adults, one suffers from CKD (4.7% in men and 5.8% in women) [1]. AKI is a rapid loss of kidney function triggered by dehydration, infection, or drug toxicity, resulting in a reduction in urine output with increased serum creatinine and blood urea nitrogen levels [2,3]. Many causes potentiate kidney injury, including renal ischemia, hypovolemia, toxic renal injury, and sepsis [4,5]. However, therapeutic strategies often fail to restore the deteriorated kidney function and 25% of AKI patient develop CKD, with increasing mortality levels due to cardiovascular complications, especially coronary diseases [6]. Inflammation and oxidative stress are associated with the development of CKD and halt the renal repairing mechanism by the podocytes and fibroblast cells and reduce the renal blood flow via an angiopathic mechanism [7].

The abovementioned consequences initiate an epithelial-to-mesenchymal transition, where the population of fibroblast/myofibroblast increases and promotes renal fibrosis upstream regulation of the transforming growth factor-beta (TGF-β)/fibrosis signaling pathway, resulting in excessive extracellular matrix protein deposition (ECM), thus causing tubulointerstitial nephritis and glomerulosclerosis [8]. Renal fibrosis leads to CKD and increases urinary albumin outputs [9]. On the other hand, there are several non-coding RNAs comprising both microRNAs and long-noncoding RNAs that potentiate or inhibit the progression of CKD; among those is mir-34a, which activates renal fibrosis via downregulating klotho (an endogenous inhibitory of the fibrosis) and induces renal tubular epithelial-to-mesenchymal transition, thus provoking renal fibrosis [10]. On the contrary, SNHG-7 long non-coding RNA is an autophagy coactivator since it acts as a sponge for mir-34a that impedes its AMPK inhibitory action [11,12]. AMPK causes a downstream regulation for the PI3K/AKT/mTOR pathway and potentially upstream regulates the expression of ULK1/2 and potentiates the self-repairing mechanism autophagy [13]. Autophagy is one of the cellular conservative mechanisms since it conserves cellular energy, recycles damaged organelles, and impedes cellular apoptosis [14].

Autophagy plays a crucial role in renal tissue cellular hemostasis and metabolism via recycling and degrading damaged proteins, organelles, and macromolecules [15]. The accumulation of toxic protein aggregates, cytoplasmic inclusion bodies, and unfolded protein responses (UPR) have been noticed in renal cells with autophagy gene knockout that aggravate the progression of the kidney injury and potentiate severe tubular cell damage and apoptosis [16]. During cellular stress and starvation, autophagy is switched on via a cellular inhibitory signal for the mammalian target of rapamycin (mTOR); autophagy-related genes (Atg) are upregulated to conserve cellular nutrient and energy via recycling damaged protein and organelles rather than the de novo synthesis of them, which requires a lot of cellular energy that in turn exhausts the starved or injured tissue [17]. The process of autophagy comprises macroautophagy, which involves the formation of autophagosomes involving several steps; the first step is the autophagosome membrane nucleation. This step is essential for autophagy induction and comprises several molecules. The most important one is Beclin-1, which interacts with Ambra-1, Rubicon, Atg14L, Bif-1, UVRAG, HMGB-1, IP3R, and other molecules to form a Beclin 1-Vps34-Vps15 core complex, the so-called phagophore. Then, the next step is autophagosome membrane elongation, which primarily mediates two ubiquitin-like conjugation systems that promote the assembly of the ATG16L complex and the processing of LC3 ((PE) phosphatidylethanolamine). The next step is the membrane maturation and closure, which is promoted by LC3, Beclin 1, the lysosomal membrane proteins LAMP-1 and LAMP-2, the GTP-binding protein RAB7, the ATPase SKD1, the cell skeleton, the pH of lysosomes, and possibly presenilin 1 (PS1). The final step is the fusion between the lysosomes and autophagosome, forming an autophagolysosome complex for recycling the damaged proteins and organelles, conserving the cellular energy [18].

Current therapeutic regimens for CKD only slow the progression of renal impairment and its complications; these regimens include hemodialysis, managing anemia, and kidney transplantation in severe cases [19]. Moreover, kidney transplantation has several limitations, such as lacking live and cadaveric donors and a higher incidence of infection and cancers due to the long-term immunosuppressive regimens [20]. Thus, the regeneration of the deteriorated kidney represents a future promising therapeutic strategy for CKD; one of these regenerative strategies is mesenchymal stem cells transplantation (MSCs). Over the last decades, MSCs’ implication displayed a promising aid in several chronic diseases, including CKD, due to their self-renewability and multiplicity to differentiate to many cell types of different lineage and the secretory potency of these cells for several soluble trophic and growth factors that induce the internal repairing mechanism via activating the internal tissue stem cell niches through a paracrine effect [21]. However, several limitations impose their therapeutic applications, such as pulmonary trapping of MSCs after intravenous administration that limits their tissue availability, the difficulty in obtaining and producing consistent sources of MSCs with stable phenotypes, and the risk of forming ectopic tissue, granuloma, and tumors in several body organs [22,23].

Interestingly, one of MSCs’ regenerative mechanisms is secreting extracellular vesicles comprising many trophic factors and genetic materials that coordinate repairing mechanisms and cellular communications [24]. Among the different types of mesenchymal stem cells, breast milk-derived mesenchymal stem cells (Br-MSCs) are a valid option, as they can be obtained with a non-invasive method, expressing a higher level of embryonic transcription factors that ensures higher plasticity, multipotency, and regenerative capacity [25,26,27]. On the other hand, Br-MSCs could differentiate into many cell types of different lineage neurons, hepatocytes, pancreatic beta cells, osteoblasts, and adipocytes under in vitro conditions [27]. They display a higher degree of resistance to challenging conditions as they could survive GIT conditions, cross intestinal barriers, and circulate within the cardiovascular system of the infant [28,29].

Exosomes are extracellular vesicles (EVs), with a diameter of 30–150 nm formed by the fusion of multivesicular cell membranes. They can be released by almost all cells, can be transferred to target cells through cell-to-cell communication, and can perform a variety of biological functions. The current progress on exosomes in the diagnosis and treatment of diseases provides an important basis for their future application in medicine [30]. Unlike MSCs, their derived exosomes have better tissue distribution, safety, and lower tumorigenicity and immune-mediated rejection, indeed bridging regenerative regimens towards the cell-free therapy considered safe and promising reparative implication in regenerative medicine [31]. Therefore, this study was designed to explore the possible therapeutic implication of breast milk mesenchymal stem cell-derived exosomes (Br-MSCs-EXOs) in CKD and to address the underlying regenerative mechanism regarding the epigenetic regulation of the renal autophagy/fibrotic signaling pathway.

## 2. Materials and Methods

### 2.1. Experimental Animals

Eighty adult male Sprague Dawley rats 220 ± 20 gm and 6–8 weeks old were obtained from the animal house faculty of veterinary medicine, Zagazig University and were maintained under standard conditions for experimentation on rodents. Nephropathy was induced in rats via the administration of adenine (Sigma–Aldrich Chemical Co., St. Louis, MO, USA) 200 mg/kg body weight (b.wt) in 0.5% carboxymethyl cellulose (CMC) (Sigma–Aldrich Chemical Co., St. Louis, MO, USA) daily for 24 consecutive days [32].

### 2.2. Isolation and Identification of Breast Milk Mesenchymal Stem Cells (Br-MSCs)

Following informed written consents, twenty milk samples were collected from donor mothers under septic conditions from Zagazig University’s pediatric hospital. The volume of each collected breast milk sample from each mother was 15–20 mL. Breast milk mesenchymal stem cells were isolated according to the method developed by Patki et al. [33] and reported by [34,35]. After the third passage, the cells were identified with flow cytometrical analysis of positive surface markers (CD105, CD90, and CD73) using CD90FITC, CD73PE, and CD105 FITC antibodies (Minneapolis, MN, USA) and negative markers (CD34, CD45, and HLA-DR) using CD34PE, CD45FITC, and HLA-DRPE antibodies (Minneapolis, MN, USA). The Br-MSCs were labeled with PKH-26 (Sigma Aldrich, St. Louis, MO, USA) following the manufacturer’s instructions for tracking the transplanted cells [33,34].

### 2.3. Isolation, Identification, and Electron Microscopy of Breast Milk Mesenchymal Stem Cell Exosomes

Br-MSCs were maintained and incubated for about 12 h in media without fetal bovine serum (FBS) to exclude isolation of the FBS exosomes and to ensure that the obtained exosomes were the only ones secreted by the breast milk-derived mesenchymal stem cells. The cell media were centrifuged at 2000× *g* for 30 min. Then, the supernatant with the cell-free culture media was translocated to a new tube without disturbing the pellet. The required volume of cell-free culture media was added to 0.5 volumes of Total Exosome Isolation reagent (Invitrogen). After mixing by vortex, incubation was performed at 2 °C to 8 °C overnight, followed by centrifugation at 10,000× *g* for 1 h at 2 °C to 8 °C. Then, exosomes were present in the pellet at the tube bottom. The pellet was resuspended. The MSC-derived exosome cell markers were characterized using FACS, including CD9 and CD63 using CD9PE and CD63PerCP antibodies (Minneapolis, MN, USA). The exosomes were labeled with PKH-26 (Sigma Aldrich, St. Louis, MO, USA) before their administration in rats.

The ultrastructure examination of Br-MSC-derived exosomes (Br-MSCs-EXOs) was performed using transmission electron microscopy (TEM), where the exosome pellets were suspended in a mixture of 2.5% glutaraldehyde (Merck KGaA, Darmstadt, Germany), 2% paraformaldehyde (Sigma-Aldrich Co.), and 0.1 M cacodylate buffer at pH 7.4 and incubated overnight at 4 °C, then fixed with 1% osmium tetroxide (Sigma-Aldrich Co.). After that, the sample was dehydrated with ethanol and propylene oxide. Finally, the sample was embedded in agar 100 resin kit (Agar Scientific Ltd., Stansted, UK) and sectioned (50 nm). The ultra-thin section was stained using lead citrate and uranyl acetate before being investigated under transmission electron microscopy with the JEM 1200EX (Oxford, UK) [36].

### 2.4. Experimental Design

Eighty male adult Sprague Dawley rats were allocated into six equal groups, fifteen rats each. G1: control group that was injected 25 days after starting the experimental procedures with two doses of 0.25 mL DMEM intraperitoneally with seven day intervals; G2: nephropathy group (rats received an oral dose of adenine 200 mg/kg dissolved in 0.5 CMC daily for 24 consecutive days and were injected 25 days after starting the experimental procedures with two doses of 0.25 mL DMEM intraperitoneally with seven day intervals [32]; G3: nephropathy and treated at day 25 after starting the experimental procedures with two doses of 0.25 mL of conditioned media with seven day intervals (Nephro + CM); G4: nephropathy and treated at day 25 after starting the experimental procedures with two doses of 2 × 10^7^ human Br-MSCs with seven day intervals between the two doses [35] (Br-MSCs) (Nephro + MSCs); G5: nephropathy and treated at day 25 from starting the experimental procedures with two doses of Br-MSC-derived exosomes at 75 µg per rat twice with a seven day interval between the two doses [37] (Nephro + Br-MSCs-EXOs); G6: nephropathy and treated at day 25 from starting experimental procedures with two doses of Br-MSCs 2 × 10^7^ and Br-MSCs-EXOs 75 µg with a seven day interval between two doses (the exosomes were injected firstly and Br-MSCs were infused 3 h post-exosome administration). After 12 weeks, blood samples from median eye canthus were collected. Then, all rats were euthanized. Kidney tissues were perfused with phosphate buffered saline (PBS) and rapidly divided into three parts: first part, 50 mg from a cross section of the kidney, including cortex and medulla, was collected rapidly on 1 mL Qiazol (Qiagen, Germany) and stored at −80 °C for further use in total RNA extraction; second part was wrapped in aluminum foil and stored at −80 °C for oxidant/antioxidant activity; third portion was maintained in 10% neutral buffered formalin for histopathological/immunohistochemical studies.

### 2.5. Administration of Br-MSCs and Br-MSCs-EXOs

Br-MSCs were given at a dose of 2 × 10^7^ dissolved in 0.25 mL of serum-free DMEM twice (with a seven day interval) intraperitoneally to avoid pulmonary trapping of the cells [34,35] in the Nephro + Br-MSCs and Nephro + Br-MSCs + EXOs groups. Br-MSCs-EXOs were administered at a dose of 75 µg dissolved in 0.25 mL of serum-free DMEM per rat intraperitoneally twice, with a seven days interval [37] in the Nephro + EXOs and Nephro + Br-MSCs + EXOs groups. Conditioned media is a serum-free media used for exosome isolation and was administered in the Nephro + CM group 0.25 mL intraperitoneally twice at a seven day interval. The Nephro and the control group rats were administered the same volume of intraperitoneal DMEM twice at a seven day interval.

### 2.6. Biochemical Analysis

Blood urea nitrogen, serum uric acid, and serum creatinine were measured according to the manufacturer’s instructions (SPINREACT, Gerona, Spain). Additionally, to determine the oxidant/antioxidant activity, the renal tissue was homogenized and centrifuged at 3000 rpm for 10 min; the supernatant was stored at −80 until used. Following manufacturer’s instructions, renal malondialdehyde (MDA), reduced glutathione (GSH), catalase (CAT), and superoxide dismutase (SOD) were measured using the sandwich ELISA method (MyBioSource, San Diego, CA, USA) and total antioxidant capacity (TAC) was assayed using a colorimetric assay kit (ABTS, Enzyme method, MyBioSource, San Diego, CA, USA).

### 2.7. Real-Time PCR

Total RNA was isolated from mononuclear cell layer (MNC), Br-MSCs-EXOs, Br-MSCs, and kidney tissue using qiazol (Qiagen, Hilden, Germany). The total extracted RNA concentration and quality was determined using NanoDrop^®^ ND-1000 UV—Vis spectrophotometer (thermos scientific, Waltham, MA, USA). Reverse transcription for mir-34a, mir-29b, mir-181, Let-7b, and U6 were performed with the sequence of the stem-loop primer listed in Table 1 using a miScript II reverse transcription kit (Qiagen, Santa Clarita, CA, USA) following the supplier’s instructions. The primers for microRNA were designed using online software http://www.srnaprimerdb.com (accessed on 15 February 2022) from the microRNA mature sequence collected from the database for the miRNA https://www.mirbase.org/ (accessed on 15 February 2022). On the other hand, the mRNA was reverse-transcribed with a high-capacity reverse transcriptase kit (Applied Biosystem, Foster City, CS, USA). Finally, the produced cDNA was diluted at 1–5, aliquoted, and stored at −20 °C for the gene expression study. The real-time PCR reaction was conducted in a total reaction volume of 20 µL, 10 µL TOPreal™ qPCR 2X PreMIX, SYBR Green with low ROX, (Enzynomics, Daejeon, Republic of Korea), 1 µL of forward and reverse primer supplied by (Sangon Biotech, Beijing, China) Table 1, and nuclease-free water up to 20 µL, as previously reported in [34,35,38]. The expression levels of mir-34a, mir-29b, mir-181, and Let-7b were normalized to U6 as the miRNA reference gene and mRNA was normalized to Gapdh. The relative expressions of both miRNA and mRNA were performed with 2^−ΔΔCT^ [39].

### 2.8. Histopathological Examination

Five rats per treatment group had their kidney tissue fixed in 10% neutral formalin at room temperature for 48–72 h before being dehydrated in a series of increasingly stronger alcohols, immersed in benzene for 20 min, embedded in paraffin, and cut into 5 m thick slices. Sections were deparaffinized in xylene and stained with hematoxylin and eosin (HE) to evaluate histopathological changes or with Masson’s trichrome (MTC) to identify regions of interstitial fibrosis using a Nex ES Special Stainer (Ventana Medical Systems, Roche Diagnostics Solutions, Tucson, AZ, USA). The slices were then examined under a light microscope [42].

### 2.9. Immunohistochemical Analysis

The avidin–biotin–peroxidase complex technique was used for the immunohistochemical staining of tissues. Briefly, formalin-fixed sections were heated for 20 min in a 10 mM citrate buffer (pH 6.0), cooled for 20 min at room temperature, and then subjected to primary antibodies at 4 °C overnight. Beclin-1 antibody, (rabbit polyclonal 1:500, cat. no. ab217179, Abcam, Cambridge, UK), LC3-III antibody (rabbit polyclonal, 1:4000; cat. no. NB600-1384, Novus Biologicals, Cambridge, UK), and p62/SQSTM1 antibody, (mouse monoclonal, 1:40,000, cat. no. WH0008878M1, Sigma-Aldrich, Buchs, Switzerland) were used. Phosphate-buffered saline (PBS) was used for 2X washing of the stained section with the primary antibody for 5 min. Sections were successively treated for 15 min with the peroxidase-conjugated streptavidin (1:3000 in PBS) and the appropriate secondary antibody. Visualization of immunolabeling was performed using 0.02% 3,3′ diaminobenzidine tetrahydrochloride. Hematoxylin counterstained sections were dried in gradient ethanol and mounted in Canada balsam.

### 2.10. Statistical Analysis

The mean ± standard error mean (S.E.M) was used to describe continuous variables. The normality of the data was checked with a Shapiro–Wilk test. In homogeneous data, one-way analysis of variance (ANOVA) was used, followed by Tukey’s honest significant difference test for multiple group comparison using SPSS (Version 20; SPSS Inc., Chicago, IL, USA); a *p* value less than 0.05 (*p* < 0.05) was considered statistically significant.

## 3. Results

### 3.1. Identification and Homing of Br-MSCs and Their Derived Exosomes

The isolated cells from breast milk showed a typical fibroblastic appearance; they could adhere to the bottom of the culture vessels (Figure 1A). On the other hand, these cells were positive for CD105, CD90, and CD73 and negative for CD34, CD45, and HLA-DR (Figure 1D–I). Moreover, their derived exosomes showed a positive reaction with CD9 and CD63 when examined with flow cytometry and appeared as spherical bodies under TEM (Figure 2A–C).

Interestingly, the transplanted Br-MSCs and/or their derived exosomes could home injured renal tissue; this was detected by the expression of human GAPDH within the “rats” renal tissue at 12 weeks (Figure 2D) and the red fluorescence emission that appeared during the examination of Br-MSCs (Figure 1B), EXOs (Figure 2F), and Br-MSCs and EXOs (Figure 1C)-treated groups’ renal tissue by fluorescent microscopy on the 3rd day post-second transplantation. Interestingly, the Br-MSCs showed a higher expression level of embryonic transcription factors Nanog, OCT4, OCT3, and SOX2 than the mononuclear cell layer isolated from human peripheral blood that comprised mature adult mesenchymal stem cells (Figure 2E).

### 3.2. Br-MSCs and/or Their Derived EXOs Improved Renal Function Tests

The nephropathic and CM-treated rats showed a significant (*p* < 0.001) elevation in the mean value of the serum uric acid, creatinine, and urea compared with control rats (Figure 3A–C). However, Br-MSCs + EXO, EXOs, and Br-MSC-treated groups provoked a significant (*p* < 0.001) reduction in the abovementioned parameters in orders compared with nephropathy and CM-treated groups (Figure 3A–C).

### 3.3. Br-MSCs and/or Their Derived EXOs’ Impact on Renal Oxidant/Antioxidant Markers

The nephropathy and CM-treated groups had a significant (*p* < 0.001) increase in the mean value of lipid peroxidation markers (MDA) and a decrease in the mean value of the antioxidant marker SOD, GPx, and CAT compared with the control group (Figure 4A–D). On the contrary, transplantation of Br-MSCs + EXO, EXOs, and Br-MSCs provoked a significant (*p* < 0.001) decline in the mean value of lipid peroxidation markers (MDA) and an elevation in the mean value of the antioxidant marker SOD, GPx, and CAT compared with nephropathic rats, respectively, compared with nephropathy and CM-treated groups (Figure 4A–D).

### 3.4. Br-MSCs and/or Their Derived EXOs’ Impact on Renal Expression of mir-34a/SNHG-7 /AMPK/ULK-1–AKT/mTOR Signaling Pathway

There was significant (*p* < 0.001) upregulation in the mean fold change for the relative expression of mir-34a, mTOR, and AKT and downregulation in the mean fold change for the relative expression of SNHG-7, AMPK, and ULK-1 in the nephropathy and CM-treated rats compared with control ones (Figure 5A–F). In contrast, Br-MSCs + EXO, EXOs, and Br-MSCs’ transplantation caused a significant (*p* < 0.001) downstream regulation of mir-34a, mTOR, and AKT expression, and upstream regulation in SNHG-7, AMPK, and ULK-1 expression, respectively, compared with nephropathy and CM-treated groups (Figure 5A–F).

### 3.5. Br-MSCs and/or Their Derived EXOs Affect Renal Histopathology

Examination of H&E-stained sections of the renal cortex of the control group showed entirely normal histological features (Figure 6A). However, kidneys of nephropathy and nephropathy + CM groups showed nearly similar histological features characterized by marked tissue injury in the form of shrinkage of glomerular capillary loops; a parietal layer of Bowman’s capsule was irregular with wide Bowman’s space. Proximal and distal tubules were hardly distinguished, with apparent wide spaces between them. The tubular epithelium exhibited vacuolation and desquamation into the lumina of the tubules. The desquamated cells partially obliterated the lumina of some tubules. Moreover, some tubules showed dark shrunken pyknotic nuclei. Furthermore, foci of inflammatory cell infiltration, extravasation, and congested thick peritubular vessels were demonstrated in between the cortical tubules (Figure 6B,C).

For the nephropathy + MSCs group, moderate restoration of histopathological renal architecture was detected in some sections, while others showed persistent injury of the renal tissue in the form of shrunken glomeruli, wide Bowman’s space, and congested thick-walled vessels. Wide spaces can be detected between the affected tubules with dark pyknotic nuclei and some exfoliated cells. Inflammatory cell infiltration can be seen in the interstitium (Figure 6D). In contrast, administration of EXOs in the nephropathy + EXOs group revealed marked improvement of the histopathological renal architecture, which was more obvious than in the nephropathy + MSCs group. However, few histopathological changes indicating disruption of the renal tissue were still detected in a few sections. These changes were in the form of pyknotic nuclei in a few tubular epithelial linings, while other nuclei were vesicular; few exfoliated tubular cells and persistent vast space between affected tubules were apparent. In addition, a few inflammatory cell infiltrations and congested blood capillaries can be seen (Figure 6E). On the other hand, cortical renal sections of the nephropathy + MSCs + EXOs group showed a histological profile comparable to the control group. Proximal and distal convoluted tubules could be distinguished. Proximal tubules showed narrow star-shaped lumina, while the lumina of distal tubules were wider. The tubules were lined with deep acidophilic cuboidal cells, with vesicular nuclei, except for a few distal tubules showing darkly stained nuclei. Renal corpuscles showed intact structure (Figure 6F).

### 3.6. Br-MSCs and/or Their Derived EXOs Affect Renal Beclin-1 mRNA and Protein Expression

There was more significant (*p* < 0.001) downregulation in both mRNA and protein expression of renal Beclin-1 either in nephropathic rats or CM-treated ones than the control rats (Figure 7A–D,H). On the contrary, Br-MSCs + EXO, EXOs, and Br-MSC-treated groups showed more significant upregulation in the mean value of renal Beclin-1 either gene or protein expression than the nephropathy and CM-treated groups, harmoniously (Figure 7A,E–H).

### 3.7. Br-MSCs and/or Their Derived EXOs Affect Renal LC3-II mRNA and Protein Expression

The nephropathy and CM-treated group revealed a significant (*p* < 0.001) downregulation in both mRNA and protein expression of renal LC3-II compared with the control group (Figure 8A–D,H). On the opposite side, Br-MSCs + EXO, EXOs, and Br-MSCs’ transplantation induced a significant upregulation in the mean value of gene and protein expression of renal LC3-II compared with the nephropathy and CM-treated groups correspondingly (Figure 8A,E–H).

### 3.8. Br-MSCs and/or Their Derived EXOs Affect Renal P62 mRNA and Protein Expression

There was a more significant (*p* < 0.001) upregulation in mRNA and protein expression of renal P62 of nephropathic and CM-treated groups than the control rats (Figure 9A–D,H). However, Br-MSCs + EXO, EXOs, and Br-MSCs’ administration showed significant downregulation in the mean value of renal P62 mRNA and protein expression compared with the nephropathy and CM-treated groups, respectively (Figure 9A,E–H).

### 3.9. Expression Pattern of Antifibrotic microRNAs and Effect of Br-MSCs and/or Their Derived EXOs on Their Expression

Both Br-MSCs and EXOs expressed the antifibrotic microRNAs, mir-29b, mir-181, and Let-7b; however, the expression level in exosomes was significantly (*p* < 0.001) higher than Br-MSCs (Figure 10A). Moreover, the nephropathy group showed a significant (*p* < 0.001) downregulation in the expression of mir-29b, mir-181, and Let-7b compared with the control group. Interestingly, administration of Br-MSCs + EXOs, EXOs, and Br-MSCs significantly (*p* < 0.001) upregulated the expression of the aforementioned antifibrotic microRNAs in order (Figure 10B–D).

### 3.10. Br-MSCs and/or Their Derived EXOs’ Effect on the Renal TGF-β/Smad/Fibrotic Signaling Pathway

There was a significant (*p* < 0.001) upregulation in the mean fold change of TGF-β, Smad-3, and coli-1 relative mRNA expression and downregulation in the mean fold change of Smad-7 relative mRNA expression. In addition, there was a more significant increase in the mean area % of collagen deposition of the nephropathy and CM-treated groups than the control one (Figure 11A–G,K). On the contrary, Br-MSCs, EXOs, and Br-MSCs + EXOs’ administration improved the parameters mentioned above significantly (*p* < 0.001) in rank order of Br-MSCs + EXO, then EXOs followed by Br-MSCs (Figure 11A–D,E–K).

## 4. Discussion

In the last few decades, kidney diseases have shown rising prevalence along with limited therapeutic aids. Moreover, almost all of these therapeutic approaches slow the diseases’ progression and delay the associated complications rather than reversing them [43]. Mesenchymal stem cells (MSCs) are promising as curative strategies that impose and reverse the progression of kidney disease via several repairing mechanisms; however, their application has many limitations, such as limited tissue bioavailability due to pulmonary trapping after intravenous administration, the tendency to form tumors in a different body organ, and the difficulty to obtain a stable source of cells with the same phenotype [22,23]. On the other hand, mesenchymal stem cells secrete exosomal vesicles that contain many soluble trophic and growth factors and genetic material such as mRNA and miRNA that could ameliorate progressive tissue damage and exert a tissue reparative mechanism [44]. This study showed that the isolated cells showed a typical fibroblastic appearance with a phenotypic characterization positive for CD90, CD105, and CD73, also negative for CD45, CD34, and HLA-DR, which illustrated that the isolated cells displayed typical characteristics of the mesenchymal stem cells [34,35,45]. Conversely, the isolated exosomes were positive for the exosomal surface markers CD9 and CD63, which are tetraspanin proteins that distinguish the exosomes from the microvesicles. Moreover, they appeared with TEM as round vesicles 100–200 nm in diameter, clearly defining that the isolated vesicles were exosomes [46,47]. The intraperitoneal route was selected to avoid pulmonary trapping and apoptosis of the infused BM-MSCs that were experienced with the intravenous route that limited tissue bio-distribution and availability of the transplanted MSCs [48]. The intraperitoneal route for MSCs infusion was validated in several previous studies [34,35,38]. Regarding the multiplicity of the breast milk-derived mesenchymal stem cells (Br-MSCs), the MNC layer was used as a representative control that comprised adult mature mesenchymal stem cells representing an easily obtainable source without major invasion since it was obtained with Ficoll density gradient separation of peripheral blood sample and comprised all mononuclear cells, including peripheral blood adult mature mesenchymal stem cells. This was carried out to evaluate the embryonic potency of the breast milk-derived MSCs over the adult one that ensured higher multiplicity and regenerative capacity of the breast milk-derived mesenchymal stem cells [49].

AKI and CKD genesis and progression both entail oxidative stress [50]. Oxidative stress causes renal inflammation, tubular fibrosis, and tubular epithelial cell death, which promotes the course of KD [51,52]. The current study showed that the nephropathy group revealed a significant elevation in kidney function tests and lipid peroxidation markers and a significant reduction in CAT, SOD, GSH, and TAC activities following [53]. Interestingly, the administration of Br-MSCs + EXOs, EXOs, and Br-MSCs improved renal function tests and oxidant/antioxidant status, respectively. Several studies reported that MSCs [54,55] and exosomes [56] have been used as an antioxidant therapeutic medication for KD. The antioxidant properties of both MSCs and EXOs upregulate the expression of Calbindin-1, which sequesters excess calcium and reduces reactive oxygen species (ROS) generation and reduces apoptosis [57]. Also, MSCs and their derived exosomes could abrogate the oxidative stress via activation of the Nrf-2/Keap-1 pathway and improve mitochondrial function by a paracrine effect due to the secretion of several soluble trophic and growth factors (HO-1, VEGF, IGF, and IDO) [58].

Furthermore, several microRNAs have been implicated in developing renal fibrosis. Among them is mir-34a via downregulating klotho, an endogenous inhibitor of the renal fibrosis and activating TGF-β pathway that enhances EMT promoting the transition of the fibroblast into myofibroblast and increasing extracellular matrix deposition [10]. Also, mir-34a has been reported to decrease the expression of AMPK and increase the expression of mTOR that potentiate EMT and increase deposition of the fibrotic proteins [59,60]. The result of the current work is in accordance with the abovementioned, as the nephropathy group showed a sharp elevation in the expression of the renal mir-34a, AKT, and mTOR with marked downregulation in the expression of ULK-1 and AMPK. However, Br-MSCs-EXOs + Br-MSCs, EXOs, and Br-MSCs’ transplantation significantly downregulated the expression of mir-34a, AKT, and mTOR and upregulated the expression of AMPK and ULK-1, respectively, which could be attributed to the long non-coding RNA SNHG-7 that acted as a sponge for mir-34a and imposed its AMPK inhibitory activity. AMPK activation could inhibit AKT activation, which in turn could diminish the EMT of the fibroblast to myofibroblast and reduce extracellular matrix protein deposition [11,12].

Surprisingly, SNHG-7 long non-coding RNA activates directly activated autophagy [61,62]. Autophagy is the cell quality control and repairing system. It is responsible for recycling the damaged cellular organelles and conserving cellular energy under stress conditions that potentially coordinate with the ubiquitin system to eliminate the toxic protein that imposes cellular inflammation and apoptosis [63]. Interestingly, the outcomes of the present work are on the same ground as the consequence mentioned above, since the nephropathic group showed a significant downregulation in the expression of SNHG-7 long-noncoding RNA, Beclin-1, and LC3-II and a prominent upregulation in mir-34a and P62 expression that could actively switch off the autophagy pathway, increasing renal cellular inflammation and apoptosis [64]. On the other hand, administration of the Br-MSCs + EXOs, EXOs, and Br-MSCs caused a marked elevation in the expression of SNHG-7, AMPK, ULK-1, LC3-II, and Beclin-1 and significant downregulation in AKT, P62, and mTOR, which reflected a dynamic activation of the autophagy pathway that conserved renal cellular energy and reduced renal cell inflammation and apoptosis, which could be owed to the potency of Br-MSCs and/or their derived exosomes to upregulate the expression of the SNHG-7, which in turn upregulated the expression autophagy pathway that markedly noticed the upregulated mRNA and protein expression of Beclin-1 and LC3-II that were responsible for autophagic membrane nucleation and elongation [65,66]. Zhang et al. [67] stated that transplantation of the MSCs alleviated cardiac fibrosis via activating renal autophagy.

Since activation of autophagy aids in the generation of a local anti-inflammatory microenvironment via increasing CD4+ lymphocyte and inducing the macrophage in an anti-inflammatory phenotype and promoting expression of peroxisome proliferator–activated receptor coactivator-1α (PGC-1α)/transcription factor EB (TFEB)-mediated lysosome-autophagy that coordinate with the ubiquitin system in the degradation of the toxic protein that serves as auto-antigen [68], thus, limiting the further tissue inflammation and damage, whereas suppression of autophagy increases production of immunosuppressive prostaglandin E2 [69].

Moreover, renal interstitial fibrosis is a typical pathophysiological indicator of CKD progression that invariably progresses to end-stage kidney disease [70]. Renal interstitial fibrosis is characterized by fibroblast proliferation and an imbalance between the synthesis and degradation of ECM, confirming that EMT of renal tubular cells is a crucial process; inhibiting renal tubular EMT may be a promising strategy for the treatment of CKD [71]. In addition, many short non-coding miRNAs involved in the regulation of EMT, including mir-29b [72], mir-181 [73], and let-7b [74], exert an antifibrotic response different from the cell growth and proliferation action of hsa-let-7c-5p [75]. All the microRNAs mentioned above abrogate renal fibrosis via inhibiting EMT that burden the transition of the fibroblast into myofibroblast, thus reducing the deposition of extracellular matrix proteins as well as the reduction of renal fibrosis. Interestingly, the result of the present investigation showed that Br-MSCs and their derived exosomes expressed mir-29b, mir-181, and Let-7b with a significantly higher expression in EXOs than Br-MSCs that clearly illustrated the antifibrotic effect of the of Br-MSCs and/or their derived exosomes (observed with the administration of MSCs and/or EXOs in the CKD rat model) [24,76]. Moreover, the antifibrotic of the microRNAs mentioned above could be attributed to the downstream regulation of TGF-receptor, TGF-β, and Smad-3 and the upstream regulation of Smad-7, reducing ECM protein deposition in the renal tissue [74]. Collectively, the results of this work were in the same line as the aforesaid findings, as the nephropathy group showed a marked downregulation in the expression of the antifibrotic markers mir-29b, mir-181, Let-7b, and Smad-7 and upregulation in the profibrotic one, Smad-3, TGF-β, and Coli-1, which improved with Br-MSCs + EXOs, EOXs, and Br-MScs, respectively. Thus, these findings illustrated that the Br-MSCs and/or their derived EXOs displayed a renoprotective and repairing effect via repressing the dynamic of the renal fibrosis by secreting a group of antifibrotic microRNAs (mir-29b, mir-181, and Let-7b).

## 5. Conclusions

In conclusion, Br-MSCs and their derived exosomes have the potential to ameliorate adenine-induced nephropathy and attenuate renal fibrosis through various mechanisms, including upregulating the expression of long-noncoding RNA NBR-2 to induce renal autophagy and inhibiting EMT by secreting mir-29b, mir-181, and Let-7b to reduce extracellular matrix protein deposition in renal tissue. Notably, the study highlights cell-free therapy (exosomes)’s superiority over cell-based therapy (Br-MSCs). However, further studies are needed to validate these findings and to investigate additional underlying mechanisms. These novel therapeutic strategies hold great promise for the treatment of kidney disease and for improving patient outcomes.

## Figures and Tables

**Figure 1 pharmaceutics-15-02149-f001:**
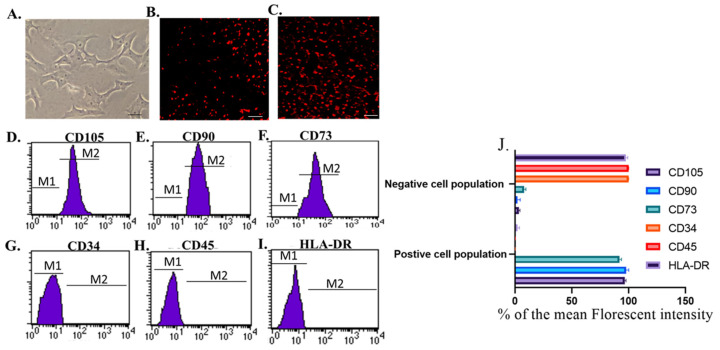
Identification and homing of Br-MSCs (**A**–**I**). (**A**) Br-MSCs cells 7th of culture, Scale bar 20 µm, (**B**) homing of Br-MSCs in rat kidney tissue of Br-MSCs treated group 3rd post-transplantation assayed by PKH-26, Scale bar 100 µm, (**C**) homing of Br-MSCs in rat kidney tissue of Br-MSCs + EXOs treated group 3rd post-transplantation assayed by PKH-26, Scale bar 100 µm, (**D**) flow cytometry of CD105 positive Br-MSCs, (**E**) flow cytometry of CD90 positive Br-MSCs, (**F**) flow cytometry of CD73 positive Br-MSCs, (**G**) flow cytometry of CD34 negative Br-MSCs, (**H**) flow cytometry of CD45 negative Br-MSCs, (**I**) flow cytometry of HLA-DR negative Br-MSCs, and (**J**) % of the mean fluorescent intensity for positive and negative cell population for CD105, CD90, CD73, CD34, CD45, and HLA-DR.

**Figure 2 pharmaceutics-15-02149-f002:**
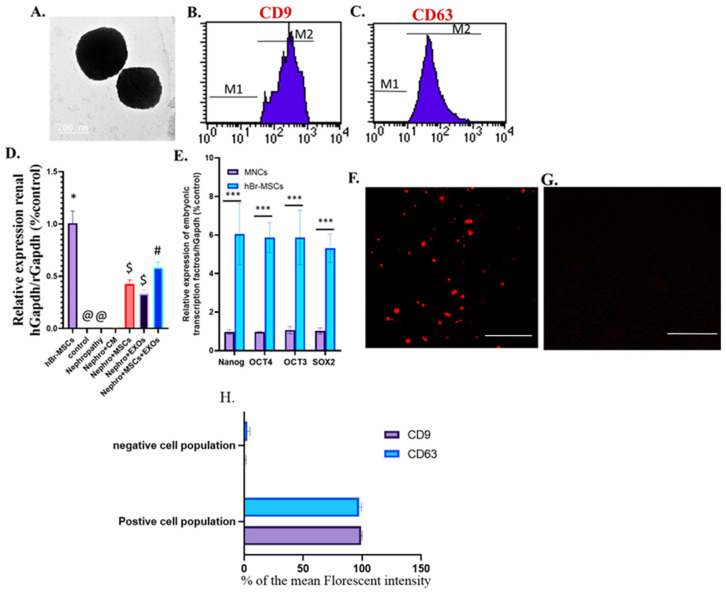
Identification and homing of Br-MSCs and exosomes and expression of the embryonic transcriptional factor of Br-MSCs (**A**–**F**). (**A**) TEM photomicrograph of exosomes; (**B**) flow cytometry of CD9 positive exosomes; (**C**) flow cytometry of CD63 positive exosomes; (**D**) detection of Br-MSCs, exosomes, Br-MSCs + EXOs in rat renal tissue at the end of the experiment via relative expression of human Gapdh mRNA at the end of the experiment after 12 weeks; (**E**) expression of Br-MSCs embryonic transcriptional factors (Nanog, OCT3/4, Sox2) compared with human mononuclear cell layer (MNC); (**F**) homing of exosomes in renal tissue of EXOs rats group 3rd day post-transplantation assayed by PKH-26, Scale bar 400 µm,; (**G**) % of the mean fluorescent intensity for positive and negative cell population for CD9 and CD63, Scale bar 400 µm, (**H**) representative photomicrograph for renal tissue of Nephro + CM group under florescent microscope showing negative florescent signal; *** *p* < 0.001, ^@,#,$,^* Means bearing different superscripts were significantly different at *p* < 0.05.

**Figure 3 pharmaceutics-15-02149-f003:**
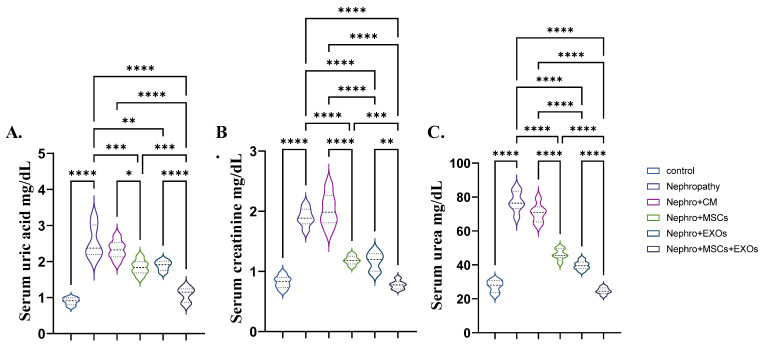
Effect of Br-MSCs and/or their derived EXOs on renal function tests (**A**–**C**). (**A**) Serum uric acid mg/dL, (**B**) serum creatinine mg/dL, and (**C**) serum urea mg/dL. Values are the mean of 6–8 rats per group ± S.E.M. * *p* < 0.05, ** *p* < 0.01, *** *p* < 0.001, **** *p* < 0.0001.

**Figure 4 pharmaceutics-15-02149-f004:**
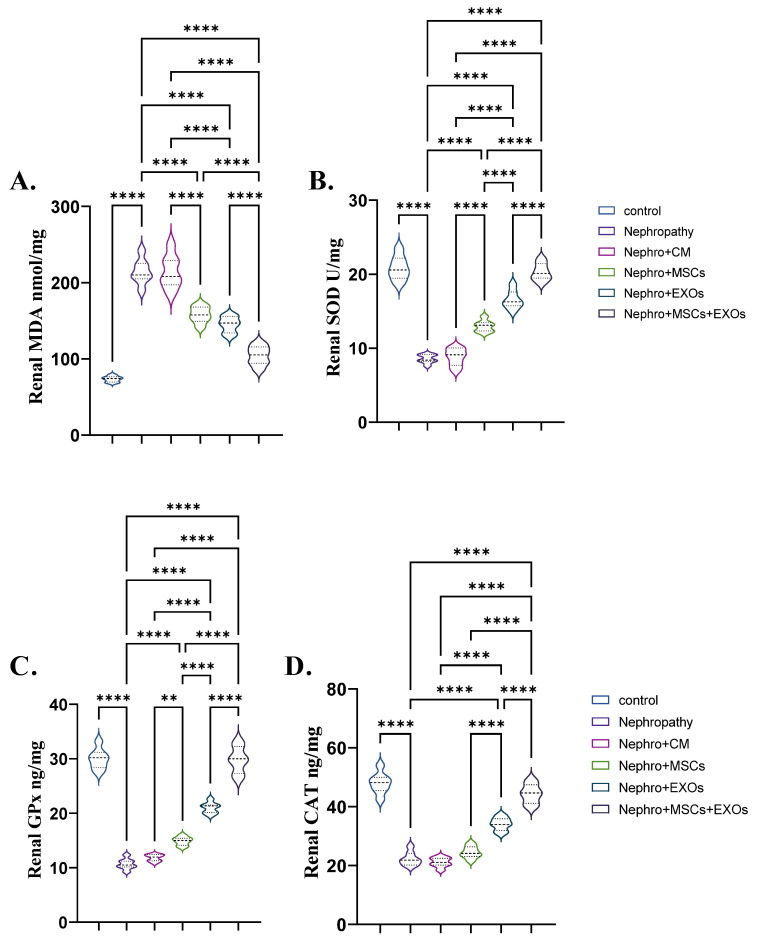
Effect of Br-MSCs and/or their derived EXOs on renal oxidant/antioxidant markers (**A**–**D**)**.** (**A**) renal MDA (nmol/mg), (**B**) renal SOD (U/mg), and (**C**,**D**) renal CAT (ng/mg). Values are the mean of (*n*) 6–8 rats per group ± S.E.M. ** *p* < 0.01, **** *p* < 0.0001.

**Figure 5 pharmaceutics-15-02149-f005:**
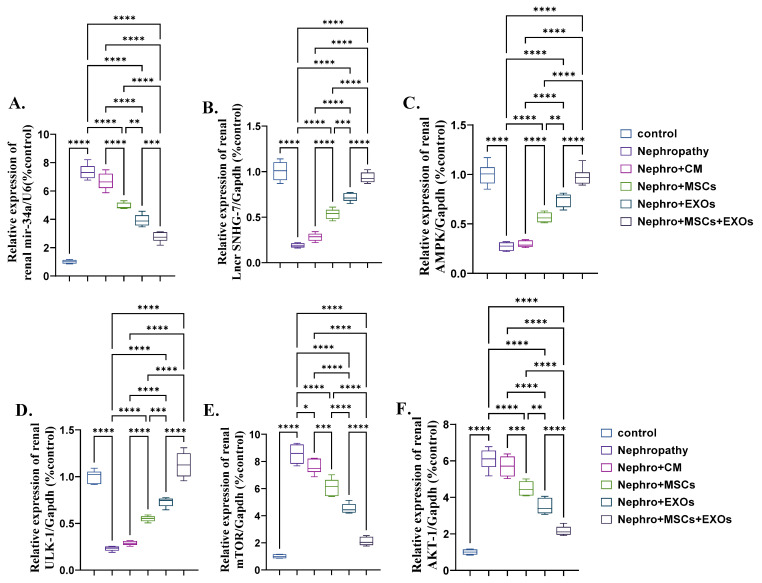
Effect of Br-MSCs and/or their derived EXOs on renal expression of mir-34a/SNHG-7/AMPK/ULK-1–AKT/mTOR signaling pathway (**A**–**F**). (**A**) Relative expression of mir-34a/U6 (% control), (**B**) relative expression of Lncr SNHG-7/Gapdh (% control), (**C**) relative expression of AMPK/Gapdh (% control), (**D**) relative expression of ULK-1/Gapdh (% control), (**E**) relative expression of mTOR/Gapdh (% control), and (**F**) relative expression of AKT-1/Gapdh (% control). Values are the mean of (*n*) 6–8 rats per group ± S.E.M. * *p* < 0.05, ** *p* < 0.01, *** *p* < 0.001, **** *p* < 0.0001.

**Figure 6 pharmaceutics-15-02149-f006:**
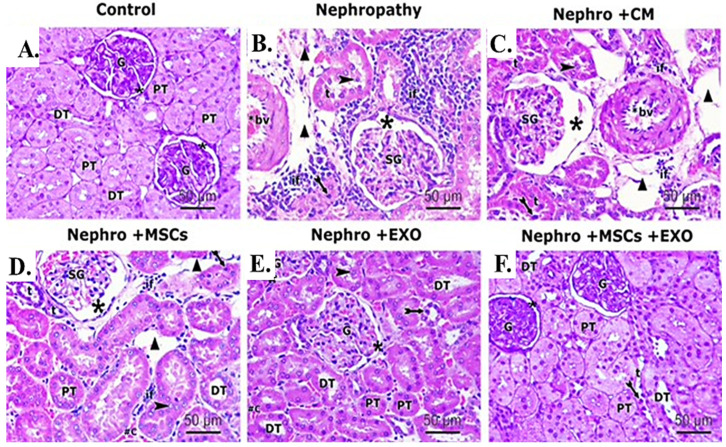
Effect of Br-MSCs and/or their derived EXOs on the renal histopathological changes (**A**–**F**). (**A**) Photomicrographs of H&E-stained sections showing histological features of control group, showing a spherical renal corpuscle lined by Bowman’s capsule, the glomerulus (G) appears as a large cellular mass surrounded by narrow Bowman’s space (star). Proximal convoluted tubules (PT) have a narrow star-shaped lumen and are lined by acidophilic cuboidal cells with an apical brush border. The distal convoluted tubules (DT) have a wider lumen and are lined by more cells. (**B**) Photomicrographs of H&E-stained sections showing histological features of the nephropathy group showing marked distorted cortical structures and features of tubule degeneration. The lining epithelium of some tubules (t) is sloughed into their lumina (arrowhead). Other tubules showed vacuolation of the lining epithelium with pyknotic nuclei (bifid arrow). Shrunken glomeruli (SG), wide Bowman’s space (star), and wide spaces between tubules (triangle) are seen. Heavy inflammatory cell infiltration (if) and thick-walled blood vessels (bv) can be noticed between the tubules. (**C**) Photomicrographs of H&E-stained sections showing histological features of nephropathy+ CM group, showing nearly similar histological findings as in nephropathy group in the form of exfoliated tubular cells (arrowhead), pyknotic nuclei (bifid arrow), shrunken glomeruli (SG), and wide Bowman’s space (star). Also, foci of inflammatory cell infiltration (if), wide spaces (triangle) between the tubules (t), and thick-walled blood vessel (bv) can be noticed. (**D**) Photomicrographs of H&E-stained sections showing histological features of nephropathy + MSCs group showing some normal cortical tubules having wide lumina (DT) and star-shaped lumina (PT). However, the lining epithelium of affected tubules showed pyknotic nuclei (bifid arrow) and the lumina of some tubules (t) showed exfoliated epithelium (arrowhead). Shrunken glomeruli (SG), wide Bowman’s space (star), wide spaces between tubules (triangle), and inflammatory cell infiltration (if) can be seen in the interstitium. (**E**) Photomicrographs of H&E stained sections showing histological features of nephropathy + EXOs group showing marked restoration of normal cortical tubules architectures that appeared with wide lumina (DT), star-shaped lumina (PT), and normal glomerular structure (G), surrounded by narrow Bowman’s space (star). However, the lining epithelium of few affected tubules showed dark nuclei (bifid arrow). (**F**) Photomicrographs of H&E-stained sections showing histological features of nephropathy + MSCs + EXOs group, showing a histological profile comparable to the control group in the form of normal glomerular structure (G), surrounded by narrow Bowman’s space (star). Proximal convoluted tubules (PT) lined by tubular cells have a star-shaped lumen and vesicular nuclei. Distal convoluted tubules (DT) have wider lumen. Only a few distal tubules (t) showed dark nuclei (bifid arrow) (H&EX 400, Scale bar = 50 μm). *n* = 5–6 rats per group.

**Figure 7 pharmaceutics-15-02149-f007:**
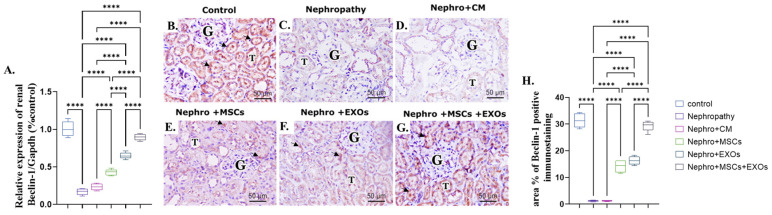
Effect of Br-MSCs and/or their derived EXOs on renal Beclin-1 mRNA and protein expression (**A**–**H**). (**A**) Relative expression of Beclin-1/Gapdh (% control); (**B**) photomicrographs of anti-Beclin immune-stained sections of the renal cortical structures in control group showed moderate positive brown cytoplasmic immunostaining (arrow head) in the epithelial cells of the renal tubules (T) and glomerular endothelial cells (G); (**C**,**D**) photomicrographs of anti-Beclin immune-stained sections of the renal cortical structures in nephropathy and nephropathy + CM groups showed marked reduction in cytoplasmic immunoreactivity in the epithelial cells of the renal tubules (T) and glomerular cells (G); (**E**,**F**) photomicrographs of anti-Beclin immune-stained sections of the renal cortical structures in nephropathy + MSCs and nephropathy + EXOs groups showed marked increase in the immune reaction (arrow head) in the renal tubular cells (T) and the glomerular cells (G); (**G**) photomicrographs of anti-Beclin immune-stained sections of the renal cortical structures in nephropathy + MSCs + EXOs group showed strong immune reaction similar to that of control group (arrow head) in the epithelial cells of the renal tubules (T) and the glomerular cells (G); (**H**) area % of Beclin-1 positive immunostaining. Values are the mean of (*n*) 6–8 rats per group ± S.E.M. **** *p* < 0.0001. (Beclin immunoperoxidase stain counterstained with H.; X400, Scale bar = 50 μm).

**Figure 8 pharmaceutics-15-02149-f008:**
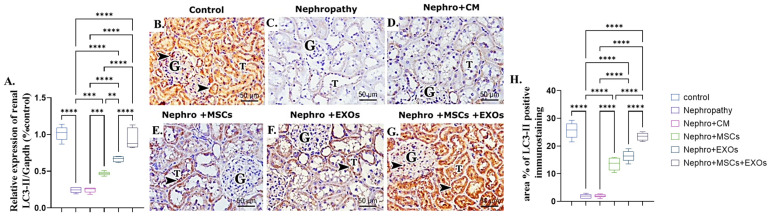
Effect of Br-MSCs and/or their derived EXOs on renal LC3-II mRNA and protein expression (**A**–**H**). (**A**) Relative expression of Beclin-1/Gapdh (% control), (**B**) photomicrographs of anti-LC3 immune-stained sections of the renal cortical structures in control group showed strong positive brown cytoplasmic immunostaining (arrow head) in the epithelial cells of the renal tubules (T) and glomerular endothelial cells (G), (**C**,**D**) photomicrographs of anti-LC3 immune-stained sections of the renal cortical structures in nephropathy and nephropathy + CM groups showed marked reduction in cytoplasmic immunoreactivity in the epithelial cells of the renal tubules (T) and glomerular cells (G), (**E**,**F**) photomicrographs of anti-LC3 immune-stained sections of the renal cortical structures in nephropathy + MSCs and nephropathy + EXOs groups showed marked increase in the immune reaction (arrow head) in the renal tubular cells (T) and the glomerular cells (G), (**G**) photomicrographs of anti-LC3 immune-stained sections of the renal cortical structures in nephropathy + MSCs + EXOs group showed strong immune reaction similar to that of control group (arrow head) in the epithelial cells of the renal tubules (T) and the glomerular cells (G), and (**H**) area % of LC3-II positive immunostaining. Values are the mean of (*n*) 6–8 rats per group ± S.E.M. ** *p* < 0.01, *** *p* < 0.001 **** *p* < 0.0001. (LC3 immunoperoxidase stain counterstained with H.; X400, Scale bar = 50 μm.)

**Figure 9 pharmaceutics-15-02149-f009:**
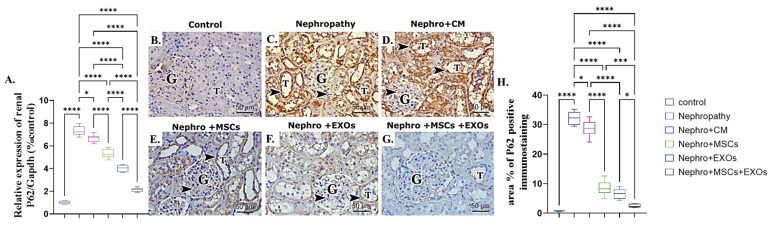
Effect of Br-MSCs and/or their derived EXOs on renal P62 mRNA and protein expression (**A**–**H**). (**A**) Relative expression of P62/Gapdh (% control), (**B**) photomicrographs of anti-P62 immune-stained sections of the renal cortical structures in control group showed negative cytoplasmic immunoreaction in the epithelial cells of the renal tubules (T) and very faint reaction in the glomerular cells (G), (**C**,**D**) photomicrographs of anti-P62 immune-stained sections of the renal cortical structures in nephropathy and nephropathy + CM groups showed strong positive reaction (dark brown color (arrow head)) in the epithelial cells of the renal tubules (T) and the glomerular cells (G), (**E**,**F**) photomicrographs of anti-P62 immune-stained sections of the renal cortical structures in nephropathy + MSCs and nephropathy + EXOs groups showed a reduction in the immune reaction (arrow head) in the renal tubular cells (T) and the glomerular cells (G), (**G**) photomicrographs of anti-P62 immune-stained sections of the renal cortical structures in nephropathy + MSCs + EXOs group showed very weak immune reaction in the epithelial cells of the renal tubules (T) and the glomerular cells (G), (**H**) area % of P62 positive immunostaining. Values are the mean of (*n*) 6–8 rats per group ± S.E.M. * *p* < 0.05, *** *p* < 0.001**** *p* < 0.0001. (P62 immunoperoxidase stain counterstained with H.; X400, Scale bar = 50 μm).

**Figure 10 pharmaceutics-15-02149-f010:**
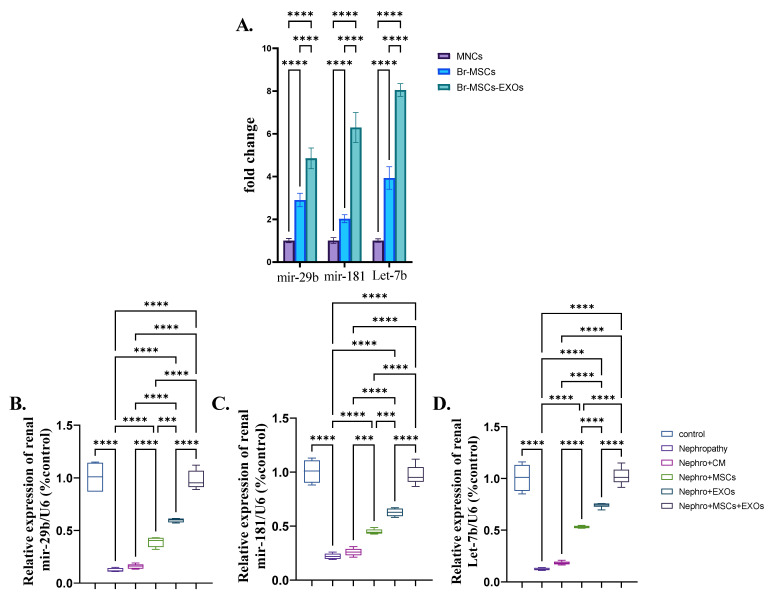
Expression pattern of antifibrotic microRNAs and effect of Br-MSCs and/or their derived EXOs on their expression (**A**–**D**). (**A**) Expression pattern of mir-29b, mir-181, and let-7b in Br-MSCs and their derived exosomes, (**B**) relative expression of renal mir-29b, (**C**) relative expression of renal mir-181, and (**D**) relative expression of renal Let-7b. Values are the means of (*n*) 6–8 rats per group ± S.E.M. *** *p* < 0.001, **** *p* < 0.0001.

**Figure 11 pharmaceutics-15-02149-f011:**
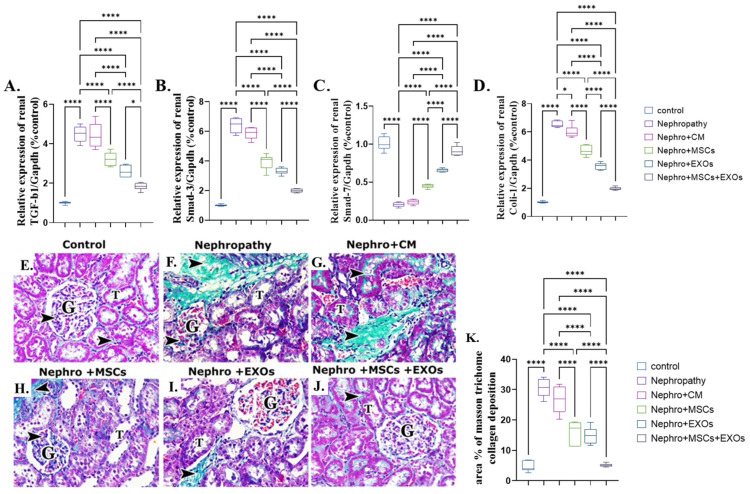
Effect of Br-MSCs and/or their derived EXOs on the renal TGF-β/Smad/fibrotic signaling pathway (**A**–**K**). (**A**) Relative expression of TGF-β/Gapdh (% control), (**B**) relative expression of Smad-3/Gapdh (% control), (**C)** relative expression of Smad-7/Gapdh (% control), (**D**) relative expression of Coli-1/Gapdh (% control), (**E**) photomicrographs of Masson trichrome-stained sections showing collagen deposition in between cortical structures (arrow head) in control group showed minimal collagen fibers among the glomerular capillaries (G) and in the interstitial in between the renal tubules (T), (**F**,**G**) photomicrographs of Masson trichrome-stained sections showing collagen deposition in between cortical structures (arrow head) in nephropathy and nephropathy + CM groups showed many collagen fibers among the glomerular capillaries (G) and in the interstitial in between the renal tubules (T), (**H**,**I**) photomicrographs of Masson trichrome-stained sections showing collagen deposition in between cortical structures (arrow head) in nephropathy + MSCs and nephropathy + EXOs groups showed a reduction in the amount of collagen fibers among the glomerular capillaries (G) and in the interstitial in between the renal tubules (T), (**J**) photomicrographs of Masson trichrome-stained sections showing collagen deposition in between cortical structures (arrow head) in nephropathy + MSCs + EXOs group showed minimal collagen fibers among the glomerular capillaries (G) and in the interstitial in between the renal tubules (T), and (**K**) Masson trichrome collagen deposition area %. Values are the mean of (*n*) 6–8 rats per group ± S.E.M. * *p* < 0.05, **** *p* < 0.0001. (P62 immunoperoxidase stain counterstained with H.; X400, Scale bar = 50 μm.) (Modified Masson trichrome × 400, Scale bar = 50 μm.)

**Table 1 pharmaceutics-15-02149-t001:** Primer sequence used for real-time PCR.

Gene	Forward Primer	Reverse Primer	bp	Accession No.	References
TGF-β1	AGGGCTACCATGCCAACTTC	CCACGTAGTAGACGATGGGC	168	NM_021578.2	[40]
Smad-7	GAGTCTCGGAGGAAGAGGCT	CTGCTCGCATAAGCTGCTGG	84	NM_030858.2	[40]
Smad-3	CTGGGCAAGTTCTCCAGAGTT	AAGGGCAGGATGGACGACAT	148	NM_013095.3	[40]
Beclin-1	GAATGGAGGGGTCTAAGGCG	CTTCCTCCTGGCTCTCTCT	180	NM_001034117.1	[40]
LC-3	GAAATGGTCACCCCACGAGT	ACACAGTTTTCCCATGCCCA	147	NM_012823.2	[40]
mTOR	GCAATGGGCACGAGTTTGTT	AGTGTGTTCACCAGGCCAAA	94	NM_019906.2	[40]
P62	GGAAGCTGAAACATGGGCAC	CCAAGGGTCCACCTGAACAA	183	NM_181550.2	[40]
SNHG-7	TGGCAGTGTCTTAGCTGGTT	AACGTGCAGCACTTCTAGGG	81	NR_031850.1	
Coli-1	GCAATGCTGAATCGTCCCAC	CAGCACAGGCCCTCAAAAAC	176	NM_053304.1	[41]
AMPK	GCGTGTGAAGATCGGACACT	TGCCACTTTATGGCCTGTCA	103	NM_023991.1	
AKT-1	GAAGGAGAAGGCCACAGGTC	TTCTGCAGGACACGGTTCTC	111	NM_033230.3	
ULK-1	CGTACACTGCCTGACCTCTC	AGAGGCCTGTGTCCCAAATG	162	NM_001108341.1	
rat Gapdh	GGCACAGTCAAGGCTGAGAATG	ATGGTGGTGAAGACGCCAGTA	143	NM_017008.4	[35]
human Gapdh	GGAGTCAACGGATTTGGTCGT	ACGGTGCCATGGAATTTGC	161	NM_002046.7	[35]
mir-29b	AACACGCCTGGTTTCACATG	GTCGTATCCAGTGCAGGGT			
mir-34a	AACACGCTGGCAGTGTCTTA	GTCGTATCCAGTGCAGGGT			
mir-181	AACACGCAACATTCAACGCT	GTCGTATCCAGTGCAGGGT			
Let-7b	AACACGCTGAGGTAGTAGGTT	GTCGTATCCAGTGCAGGGT			[40]
U6	GCTCGCTTCGGCAGCACA	GAGGTATTCGCACCAGAGGA			[40]
mir-29b stem-Loop primer		GTCGTATCCAGTGCAGGGTCCGAGGTATTCGCACTGGATACGACCTAAGC			
mir-34a stem-Loop primer		GTCGTATCCAGTGCAGGGTCCGAGGTATTCGCACTGGATACGACACAACC			
mir-181 stem-Loop primer		GTCGTATCCAGTGCAGGGTCCGAGGTATTCGCACTGGATACGACACTCAC			
Let-7b stem-Loop primer		GTCGTATCCAGTGCAGGGTCCGAGGTATTCGCACTGGATACGACAACCAC			[40]
U6 stem-Loop primer		AACGCTTCACGAATTTGCGTG			[40]

## Data Availability

Data will be provided upon reasonable request.

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
