# Peer review of "Breast Milk Mesenchymal Stem Cells and/or Derived Exosomes Mitigated Adenine-Induced Nephropathy via Modulating Renal Autophagy and Fibrotic Signaling Pathways and Their Epigenetic Regulations"

_pharmaceutics, 2023, doi:10.3390/pharmaceutics15082149_

Round 1
Reviewer 1 Report
In this study, authors analyzed the therapeutic potential of the entire MSCs from breast milk and/or their exosomes in a rat model of nephropathy. They analyze renal tissue damage by histology and by different markers in serum related to renal function and in renal tissue related to oxidant status, microRNAs, and proteins related to fibrosis and autophagy.
The scientific background, research design, and presentation of the results are well described. The discussion section is well written including all the results that they analyzed, and reasonably referenced. I would like to highlight the quality of the histology images and the recent references that they used.
In this study, it is very interesting to the numerous markers related to renal function that have analysed together with the observations of their changes by the MSC and/or exosome treatment. Some of these changes have been observed in other pathologies as the authors have referenced. Additionally, the use of breast milk as the source of MSCs is very interesting although this use is not new for other models.
From a critical view, although the rationale of the study is the analysis of exosomes to compare with the entire MSCs avoiding the lung trap and the tumor genesis the authors used the intraperitoneal route of administration which is not common in clinic instead of the intravenous route of administration. They should explain it.
Major points
In the abstract section, the organ/tissue in which the different parameters were measured should appear.
· Introduction
- Authors should explain in this section the role of some of the different parameters (beclin, LC3-II) they have analyzed in the results section.
· Materials and Methods section:
- Statistical analysis, authors assume normal data. Did they analyze the normality of the data before using ANOVA test? Looking at the graphs, some data do not match the normality test. Even the authors use violin and box plot graphs that are for non-normal data. Every data should appear as points overlaying the graphs to see their variability.
· Results section:
- How do the authors explain the absence of signal in Figure 2D in Nephro+CM group? Exos are in the culture medium depending on the isolation protocol. They should explain this point.
- Differences among the treatments should be explained in the different figures since the data point out the improvement of both treatments with respect to every treatment alone.
- Title sections should be a summary of the results that they have observed not a repetition of the legend figure title.
- They have pointed out the conclusion of every result in this section.
- I suggest changing the order of the figures, figure 11 should be before Figure 6 and, consequently, the associated text.
· Discussion section
Since one of the aims of the study is to avoid lung trap by the use of exosomes, authors should comment on the use of the intraperitoneal route of administration.
Minor points
· A section of abbreviations would help the reader
· Material and methods section:
- “2.3 Section” and “2.4 Section” should be ordered before “2.2 section” in material and methods
- “2.3 section” and 2.5” section” should be mixed. They contain practically the same information
- A brief description of Br-MSCs isolation, culture, and expansion should appear in the text
- Line 152, define FBS
- Line 158, which is the medium for exosomes resuspension?
· Results section
- Isotype controls and exos or MSCs-specific markers are needed in Figure 1 (D-I) and Figure 2 (B-C)
- Tissue controls are needed in Figure 1 (B-C) and Figure 2 (F) with specific markers for kidney tissue
- Statistical analysis of Figures 2D and 2E should appear
- Authors observed higher expression of NANOG, OCT3/4, and Sox2 in Br-MSCs than in MNCs and they do not comment anything on this result.
- Authors should improve the figure graph quality (Figures 3 and 6) for the final version of the manuscript
- MNCs should appear in Figure 2 legend
- Serum should appear in figure legends 3 and 4
- Figure legends 6, 7, 8, 10, and 11 should be reduced. Figure Legend 11 contains some details that are described in the main text.
- Arrows do not appear in Figure 6
- S.E.M should be defined the first time it is written
· References should be checked
-Line 64, reference number 2 does not match the information in the previous sentence
-Line 76, reference number 8 does not match the information in the previous sentence
-Line 77, reference number 9 does not match the information in the previous sentence
-Line 92, reference number 16 is from 2012 and the authors comment on renal costs from 2009-2020 in Australia. This should be more recent and include other countries, continents…
-Line 108, check this reference, maybe here is reference number 2
-Line 491, reference number 34 does not match the information in the previous sentence
-Line 503, here authors should reference the international criteria for MSC characterization by ISCT
-Line 563, reference number 56 does not match the information in the previous sentence
-Line 569, reference number 58 does not match the information in the previous sentence
· Discussion section
- Differences between Let-7b and Let-7c-5p (reference 61) should be commented
· Typological mistakes:
- Line 121, “b. wt”?
- Line 74, a space is needed
- Line 75, the authors should define TGFbeta
- Line 130, “Nephro” instead of “Nepehro”
- Lines 136, 137 166, and 203 “°C” instead of “oC” and in line 186 is missed
- Line 145, “+ve”?
- Lines 237, “<” is needed
- -Line 249 “fluorescence” instead of “florescence”
- Line 274, a space is needed
- Line 280, “the” is duplicated.
- Line 283, “on” should be in capital letter
- Line 508, “CKD” instead of “KD”
- -Line 559, PGC-1z/TFEB should be defined
Author Response
Thank you for your valuable feedback and suggestions. We have carefully considered each of your points and made appropriate revisions to address them.
-
Major points
In the abstract section, the organ/tissue in which the different parameters were measured should appear.
Response: Thank you for your comment. The abstract section was updated.
Introduction
- Authors should explain in this section the role of some of the different parameters (beclin, LC3-II) they have analyzed in the results section.
Response: Thank you for your comment. The introduction section was updated.
Materials and Methods section:
- Statistical analysis, authors assume normal data. Did they analyze the normality of the data before using ANOVA test? Looking at the graphs, some data do not match the normality test. Even the authors use violin and box plot graphs that are for non-normal data. Every data should appear as points overlaying the graphs to see their variability.
Response: Thank you for your comment. The materials and methods section was updated. The normality of the data were checked with shapiro wilk test.
- Results section:
- How do the authors explain the absence of signal in Figure 2D in Nephro+CM group? Exos are in the culture medium depending on the isolation protocol. They should explain this point.
Response: Many thanks for your valuable comment and for focusing on such point which is controversial. The absence of the signal was detected by the fluorescent microscope for kidney sections of the nepheropathy + CM group that relieve a negative red flourscence emission of these sections and double confirmed by detecting the expression of the human Gapdh figure 2E with using highly specific Gapdh primer which highly conserved for exons closely related for human and confirmed with the blast analysis for cross reactivity that revealed no cross reactivity with rattus norvegicus or other species rather than homo sapiens. The negative single could be attributed to the very very few amount of the exosomes that present in 0.25 mL of conditioned that fail to cross and home to the kidney tissue. However, the exosomes which administered to each rats was extracted at least from 30 – 35 mL conditioned media. Additionally, we add an image for kidney tissue of nephero+CM treated group from fluorescent imaging that confer a negative signal. Fig 2G
- Differences among the treatments should be explained in the different figures since the data point out the improvement of both treatments with respect to every treatment alone.
Response: difference among treatments has been reported where *p < 0.05, **p < 0.01, ***p < 0.001, and ****p < 0.0001 defining the significant difference between groups
- Title sections should be a summary of the results that they have observed not a repetition of the legend figure title.
Response: Thank you for your suggestion.we agree that the title sections of a research paper should provide a concise summary of the main results or findings, rather than simply repeating the legend figure title, yet to avoid large titles, we have revised some of the title so that it better reflect the results of the study.
- They have pointed out the conclusion of every result in this section.
Response: Thank you for your comment.
- I suggest changing the order of the figures, figure 11 should be before Figure 6 and, consequently, the associated text.
Response: Thank you for your comment. The results section was revised accordingly.
- Discussion section
Since one of the aims of the study is to avoid lung trap by the use of exosomes, authors should comment on the use of the intraperitoneal route of administration.
Response: Thank you for your comment. The discussion section was updated accordingly. The intraperitoneal route was selected to avoid pulmonary trapping and apoptosis of the infused BM-MSCs that experienced with intravenous route that limits tissue bio-distribution and availability of the transplanted MSCs [1]. The intraperitoneal route for MSCs infusion was validated in several previous studies [2-4].
Minor points
- A section of abbreviations would help the reader
Response: Thank you for your comment. abbreviations were mentioned during the first time.
- Material and methods section:
- “2.3 Section” and “2.4 Section” should be ordered before “2.2 section” in material and methods
Response: Thank you for your comment. The materials and methods section was revised accordingly.
- “2.3 section” and 2.5” section” should be mixed. They contain practically the same information
Response: Thank you for your comment. The materials and methods section was revised accordingly.
- A brief description of Br-MSCs isolation, culture, and expansion should appear in the text
Response: Thank you for your comment. The materials and methods section was revised accordingly.
- Line 152, define FBS
Response: Thank you for your comment. The materials and methods section was revised accordingly.
- Line 158, which is the medium for exosomes resuspension?
Response : the media was defined DMEM
- Results section
- Isotype controls and exos or MSCs-specific markers are needed in Figure 1 (D-I) and Figure 2 (B-C)
Response : Many thanks for your valuable comment, isotype control in positive markers was defined as M1 cell population and in negative markers defined as M2 and the percent of positive and negative cell population was added to fig1& 2 as bars.
- Tissue controls are needed in Figure 1 (B-C) and Figure 2 (F) with specific markers for kidney tissue
Response : we provide a representative image for nephro+CM as a negative control showing non-fluroscent signal of PKH26 fig 2G. and for double confirmation we provide figure 2D for detecting homing of the infused human breast milk-derived mesenchymal stem cells using highly specific primer for human gapdh which was validated in previous works as a tool for tracking the homing capability of the human mesenchymal stem cells inside rat tissue [3,5].
- Statistical analysis of Figures 2D and 2E should appear
Response : many thanks for this valuable notification, the figure 2D and 2E were updated accordingly.
- Authors observed higher expression of NANOG, OCT3/4, and Sox2 in Br-MSCs than in MNCs and they do not comment anything on this result.
Response : many thanks for raising such point, the result section were updated accordingly.
- Authors should improve the figure graph quality (Figures 3 and 6) for the final version of the manuscript
Response : the quality was improved
- MNCs should appear in Figure 2 legend
Response : the legend was updated accordingly
- Serum should appear in figure legends 3 and 4
Response : the legend was updated accordingly
- Figure legends 6, 7, 8, 10, and 11 should be reduced. Figure Legend 11 contains some details that are described in the main text.
Response : Thank you for your feedback. Each of these figures ligands contains important data that provide essential context and explanation for interpreting the figures. Regarding Figure Legend 11, I understand your concern about potential redundancy with information presented in the main text. However, the figure legend provides additional details and clarification that may not be immediately apparent from the main text alone. Therefore, I believe it is important to include this information in the legend to ensure that readers have a comprehensive understanding of the figure. Accordingly, if you have any specific suggestions for how to improve the clarity or conciseness of the figure legends, I am happy to consider them and make appropriate revisions where necessary.
- Arrows do not appear in Figure 6
Response : the figure was updated accordingly
- S.E.M should be defined the first time it is written
Response : the materials and method subsection statistical analysis was updated accordingly
- References should be checked
-Line 64, reference number 2 does not match the information in the previous sentence
Response: the reference was updated
-Line 76, reference number 8 does not match the information in the previous sentence
Response: the reference was updated
-Line 77, reference number 9 does not match the information in the previous sentence
Response: the reference was updated
-Line 92, reference number 16 is from 2012 and the authors comment on renal costs from 2009-2020 in Australia. This should be more recent and include other countries, continents…
Response: the sentence was removed and the reference was updated
-Line 108, check this reference, maybe here is reference number 2
Response: the reference was updated
-Line 491, reference number 34 does not match the information in the previous sentence
Response: the reference was updated
-Line 503, here authors should reference the international criteria for MSC characterization by ISCT
Response: the reference was updated
-Line 563, reference number 56 does not match the information in the previous sentence
Response: the reference was updated
-Line 569, reference number 58 does not match the information in the previous sentence
Response: the reference was updated
- Discussion section
- Differences between Let-7b and Let-7c-5p (reference 61) should be commented
Response: the discussion section was updated
- Typological mistakes:
- Line 121, “b. wt”?
Response: the typographical error was revised
- Line 74, a space is needed
Response: the typographical error was revised
- Line 75, the authors should define TGFbeta
Response: the introduction section was updated accordingly
- Line 130, “Nephro” instead of “Nepehro”
Response: the typographical error was revised
- Lines 136, 137 166, and 203 “°C” instead of “oC” and in line 186 is missed
Response: the typographical error was revised
- Line 145, “+ve”?
Response: the typographical error was revised
- Lines 237, “<” is needed
Response: the typographical error was revised
- -Line 249 “fluorescence” instead of “florescence”
Response: the typographical error was revised
- Line 274, a space is needed
Response: the typographical error was revised
- Line 280, “the” is duplicated.
Response: the typographical error was revised
- Line 283, “on” should be in capital letter
Response: the typographical error was revised
- Line 508, “CKD” instead of “KD”
Response: the typographical error was revised
- -Line 559, PGC-1z/TFEB should be defined
Response: the typographical error was revised
- Fischer, U.M.; Harting, M.T.; Jimenez, F.; Monzon-Posadas, W.O.; Xue, H.; Savitz, S.I.; Laine, G.A.; Cox, C.S., Jr. Pulmonary passage is a major obstacle for intravenous stem cell delivery: the pulmonary first-pass effect. Stem cells and development 2009, 18, 683-692, doi:10.1089/scd.2008.0253.
- Khamis, T.; Abdelalim, A.F.; Saeed, A.A.; Edress, N.M.; Nafea, A.; Ebian, H.F.; Algendy, R.; Hendawy, D.M.; Arisha, A.H.; Abdallah, S.H. Breast milk MSCs upregulated β-cells PDX1, Ngn3, and PCNA expression via remodeling ER stress/inflammatory/apoptotic signaling pathways in type 1 diabetic rats. European Journal of Pharmacology 2021, 905, 174188.
- Khamis, T.; Abdelalim, A.F.; Abdallah, S.H.; Saeed, A.A.; Edress, N.M.; Arisha, A.H. Early intervention with breast milk mesenchymal stem cells attenuates the development of diabetic-induced testicular dysfunction via hypothalamic Kisspeptin/Kiss1r-GnRH/GnIH system in male rats. Biochimica et Biophysica Acta (BBA)-Molecular Basis of Disease 2020, 1866, 165577.
- Khamis, T.; Abdelalim, A.F.; Abdallah, S.H.; Saeed, A.A.; Edress, N.M.; Arisha, A.H. Breast milk MSCs transplantation attenuates male diabetic infertility via immunomodulatory mechanism in rats. Adv. Anim. Vet. Sci 2019, 7, 145-153.
- Siniscalco, D.; Giordano, C.; Galderisi, U.; Luongo, L.; de Novellis, V.; Rossi, F.; Maione, S. Long-lasting effects of human mesenchymal stem cell systemic administration on pain-like behaviors, cellular, and biomolecular modifications in neuropathic mice. Frontiers in integrative neuroscience 2011, 5, 79, doi:10.3389/fnint.2011.00079.
Reviewer 2 Report
1. line 93-94: 'higher incidence of~': It would be more beneficial if it was expressed as an exact number (ex. %).
2. line 128: Please indicate the exact time of injection. how many days after adenine injection did you inject cells and exosomes?
3. Was the same amount of PBS injected into the G1 control group and the G2 experimental group?
4. line 141: Have you received an IRB for using human-derived materials?
5. line 192: Why did you use MNC as a control?
6. Fig1. A-C: Insert Bar.
B,C: show DAPI and provide negative control as a picture
D-I: Present the negative control and indicate the marker expression rate (%) of the cells.
7. Line 256: 3rd post –transplantation à 3rd day of ~
Why was the test done after 3 days? Is it after the 1st injection or the 2nd injection?
8. Fig.2 B,C: Present the negative control and indicate the marker expression rate (%) of the cells.
E: factorsà factors
F: show DAPI and provide negative control as picture
9. How long did the injected cells survive?
10. Also, most tests were performed at 12 weeks. How long did the effect last?
11. From which part of the kidney (eg cortex? medulla?) was the RNA extracted?
12. Mark the number (n) on the figure or figure legend.
13. Did you look at the lung histology in this experiment (to check for trapping)?
Or, if possible, it would be nice to perform in vivo cell tracking.
Author Response
Thank you for your valuable feedback and suggestions. We have carefully considered each of your points and made appropriate revisions to address them.
- line 93-94: 'higher incidence of~': It would be more beneficial if it was expressed as an exact number (ex. %).
Response: Thanks for your valuable observation, amendment was done accordingly.
- line 128: Please indicate the exact time of injection. how many days after adenine injection did you inject cells and exosomes?
Response: many thanks for your valuable comment, experimental design was amended accordingly
- Was the same amount of PBS injected into the G1 control group and the G2 experimental group?
Response: thanks for raising such an important point, the experimental design was amended accordingly, with the same amount of DMEM media 0.25ml administered in control and positive control at the rate of two doses with 7 days intervals at day 25th from starting experimental procedures and 1 day after the last dose of adenine.
- line 141: Have you received an IRB for using human-derived materials?
Response: Thank you for your comment. We take the ethical considerations of our research very seriously and have followed all necessary protocols to obtain the required approvals for the use of human-derived materials in our study. Written consent from the donors was taken as per our insititutional regulations. We have obtained the appropriate Institutional Review Board (IRB) approval for the use of human samples in our research, and have complied with all relevant ethical guidelines and regulations of BUC ethics committee and this part including the experimental methods was cleared by the Institutional Animal Care and Use Committee of Badr University in Cairo (No. BUC-IACUC/VET/128/A/2022) including the clearance to use hBr-MSC.
- line 192: Why did you use MNC as a control?
Response: the MNC layer was used as a representative control that comprised adult mature mesenchymal stem cells representing an easily obtainable source without major invasion since it was obtained with ficol density gradient separation of peripheral blood sample and comprised all mononuclear cells including peripheral blood adult mature mesenchymal stem cells, this was done to evaluate the embryonic potency of the breast milk derived MSCs over the adult one that ensures higher multiplicity and regenerative capacity of the breast milk derived mesenchymal stem cells [1]. We have used the MNC in previous work as a valid control comprising adult mature mesenchymal stem cells to compare the expression of the embryonic transcription factors of adipose tissue-derived mesenchymal stem cells [2].
- Fig1. A-C: Insert Bar.
B,C: show DAPI and provide negative control as a picture
Response: we provide a representative image for nephero+CM as a negative control showing non-fluroscent signal of PKH26 fig 2G. and for double confirmation we provide figure 2D for detecting homing of the infused human breast milk-derived mesenchymal stem cells using highly specific primer for human gapdh which confirmed with blast analysis for no cross-reactivity with other species specially Rattus norvegicus and validated in our previous works and other authors work as a valid tool for tracking the homing capability of the human mesenchymal stem cells inside rat tissue [3,4].
D-I: Present the negative control and indicate the marker expression rate (%) of the cells.
Response: many thanks for your precious comment that increases the soundness of the current work, the figure1 & 2 were updated accordingly.
- Line 256: 3rd post –transplantation à 3rd day of ~
Why was the test done after 3 days? Is it after the 1st injection or the 2nd injection?
Response: many thanks for raising such vital point, the homing were done 3 days post the second transplantation, because the cells are redistributed to the injured tissue on the 3rd day post transplantation [5]. The homing of the cells was detected 3rd days post 2nd transplantation with detecting pkh26 positive cells and 12th weeks after the end of the experiment via detecting human Gapdh expression within rat kidney tissue of all experimental group avoid fading of the PKH26 according to the manufacturer instruction.
- Fig.2 B,C: Present the negative control and indicate the marker expression rate (%) of the cells.
Response: the figure was updated accordingly.
E: factorsà factors
Response: the figure was updated accordingly.
F: show DAPI and provide negative control as picture
Response: we provide a representative image for nephero+CM as a negative control showing non-fluroscent signal of PKH26 and for double confirmation we provide figure 2D for detecting homing of the infused human breast milk-derived mesenchymal stem cells using highly specific primer for human gapdh which confirmed with blast analysis for no cross-reactivity with other species specially Rattus norvegicus and validated in our previous works and other authors work as a valid tool for tracking the homing capability of the human mesenchymal stem cells inside rat tissue [3,4].
- How long did the injected cells survive?
Response: We appreciate this very important controversial point, according to our findings in the current work. We detect higher expression level of human Gapdh within the rat renal tissue of the EXO and/or Br-MSCs treated groups with negative expression in control, positive control, and nephero+CM at 12th weeks. The Human Gapdh was detected with highly specific primer for human exon that not cross-react with other species specially Rattus norvegicus that confirmed with blast analysis and validated in previous work [3,4]. Moreover, our previous work [6] showed that the transplanted cells were integrated within several tissues and transdifferentiated into specific tissue cells like neurons which confirmed by a recent study [7].
- Also, most tests were performed at 12 weeks. How long did the effect last?
Response: we appreciate this very important controversial point, according to our findings in the current work. We detect higher expression level of human Gapdh within the rat renal tissue of the EXO and/or Br-MSCs treated groups with negative expression in control, positive control, and nephero+CM at 12th weeks. The Human Gapdh was detected with highly specific primer for human exon that not cross-react with other species specially Rattus norvegicus that confirmed with blast analysis and validated in previous work [3,4]. Moreover, our previous work [6] showed that the transplanted cells were integrated within several tissues and transdifferentiated into specific tissue cells like neurons which confirmed by a recent study [7].
- From which part of the kidney (eg cortex? medulla?) was the RNA extracted?
Response: many thanks for your comment, part 50 mg from a cross section of the kidney including cortex and medulla was collected. The materials and methods section was updated accordingly.
- Mark the number (n) on the figure or figure legend.
Response: the legends were updated accordingly.
- Did you look at the lung histology in this experiment (to check for trapping)?
Or, if possible, it would be nice to perform in vivo cell tracking.
Response: Thank you for your comment. Pulmonary trapping of the cells was recorded previously with intravenous route and validated as the first-pass effect [8]. Thus, The intraperitoneal route was selected to avoid pulmonary trapping and apoptosis of the infused Br-MSCs that experienced with intravenous route that limits tissue bio-distribution and availability of the transplanted MSCs [8]. The intraperitoneal route for MSCs infusion was validated in several previous studies [4,9,10]. In our previous study [6], we assay the biodistribution of the infused breast milk mesenchymal stem cells that showed a tissue tropism to several organs as lung, liver, pancreas, brain, heart, muscles, and others…… that confirmed with [7]. However, we appreciate your suggestion for in vivo cell tracking and will consider it for future studies.
- Gil-Kulik, P.; Leśniewski, M.; Bieńko, K.; Wójcik, M.; Więckowska, M.; Przywara, D.; Petniak, A.; Kondracka, A.; Świstowska, M.; Szymanowski, R.; et al. Influence of Perinatal Factors on Gene Expression of IAPs Family and Main Factors of Pluripotency: OCT4 and SOX2 in Human Breast Milk Stem Cells—A Preliminary Report. International journal of molecular sciences 2023, 24, doi:10.3390/ijms24032476.
- Abdelbaset-Ismail, A.; Tharwat, A.; Ahmed, A.E.; Khamis, T.; Abd El-Rahim, I.H.; Alhag, S.K.; Dowidar, M.F. Transplantation of adipose-derived mesenchymal stem cells ameliorates acute hepatic injury caused by nonsteroidal anti-inflammatory drug diclofenac sodium in female rats. Biomedicine & pharmacotherapy = Biomedecine & pharmacotherapie 2022, 155, 113805, doi:10.1016/j.biopha.2022.113805.
- Siniscalco, D.; Giordano, C.; Galderisi, U.; Luongo, L.; de Novellis, V.; Rossi, F.; Maione, S. Long-lasting effects of human mesenchymal stem cell systemic administration on pain-like behaviors, cellular, and biomolecular modifications in neuropathic mice. Frontiers in integrative neuroscience 2011, 5, 79, doi:10.3389/fnint.2011.00079.
- Khamis, T.; Abdelalim, A.F.; Abdallah, S.H.; Saeed, A.A.; Edress, N.M.; Arisha, A.H. Early intervention with breast milk mesenchymal stem cells attenuates the development of diabetic-induced testicular dysfunction via hypothalamic Kisspeptin/Kiss1r-GnRH/GnIH system in male rats. Biochimica et Biophysica Acta (BBA)-Molecular Basis of Disease 2020, 1866, 165577.
- Sanchez-Diaz, M.; Quiñones-Vico, M.I.; Sanabria de la Torre, R.; Montero-Vílchez, T.; Sierra-Sánchez, A.; Molina-Leyva, A.; Arias-Santiago, S. Biodistribution of Mesenchymal Stromal Cells after Administration in Animal Models and Humans: A Systematic Review. Journal of clinical medicine 2021, 10, doi:10.3390/jcm10132925.
- Abd Allah, S.H.; Shalaby, S.M.; El-Shal, A.S.; El Nabtety, S.M.; Khamis, T.; Abd El Rhman, S.A.; Ghareb, M.A.; Kelani, H.M. Breast milk MSCs: An explanation of tissue growth and maturation of offspring. IUBMB life 2016, 68, 935-942, doi:10.1002/iub.1573.
- Aydın, M.Ş.; Yiğit, E.N.; Vatandaşlar, E.; Erdoğan, E.; Öztürk, G. Transfer and Integration of Breast Milk Stem Cells to the Brain of Suckling Pups. Scientific Reports 2018, 8, 14289, doi:10.1038/s41598-018-32715-5.
- Fischer, U.M.; Harting, M.T.; Jimenez, F.; Monzon-Posadas, W.O.; Xue, H.; Savitz, S.I.; Laine, G.A.; Cox, C.S., Jr. Pulmonary passage is a major obstacle for intravenous stem cell delivery: the pulmonary first-pass effect. Stem cells and development 2009, 18, 683-692, doi:10.1089/scd.2008.0253.
- Khamis, T.; Abdelalim, A.F.; Saeed, A.A.; Edress, N.M.; Nafea, A.; Ebian, H.F.; Algendy, R.; Hendawy, D.M.; Arisha, A.H.; Abdallah, S.H. Breast milk MSCs upregulated β-cells PDX1, Ngn3, and PCNA expression via remodeling ER stress /inflammatory /apoptotic signaling pathways in type 1 diabetic rats. European Journal of Pharmacology 2021, 905, 174188, doi:https://doi.org/10.1016/j.ejphar.2021.174188.
- Khamis, T.; Abdelalim, A.F.; Abdallah, S.H.; Saeed, A.A.; Edress, N.M.; Arisha, A.H. Breast milk MSCs transplantation attenuates male diabetic infertility via immunomodulatory mechanism in rats. Adv. Anim. Vet. Sci 2019, 7, 145-153.
Reviewer 3 Report
General Comments
Therapeutic potential of human breast milk mesenchymal stem cells
L68; “The incidence of CKD is common with hypertension, glomerulonephritis, poly-cystic kidney disease, diabetes, and familial renal failure [6].” This is an example of unnecessary text as is essentially repeats statements already made.
The manuscript does a poor job at explaining what Br-MSC are and why they were selected for the study. It is essential that you provide more information regarding these cells. 1. Cell surface marker data alone is not sufficient to define these as MSC. 2. Is the population that is obtained from breast milk a homogeneous or heterogeneous population? 3. How many of these cells are obtained from a given volume of breast milk? 4. At what stage/stages post-partum are these cells obtained and are the donors breast-feeding during the periods when the milk is harvested?
Specific Comments
L73; “epithelial-to-mesenchymal transition where the fibroblast is transformed into myofibroblast” This is not the definition of epithelial-to-mesenchymal transition!!
Why chose breast milk MSC? Do these cells meet the basic criteria for MSCs beyond the expression of surface markers that are common to many cellular populations?
L141; Twenty milk samples This is very vague. The same woman, different women, what stage post-partum? Breasting donors, volume of samples, number of cells per sample, etc. are all important issues. Also, was an IRB protocol submitted and approved? This is necessary, even with written consent.
l120; The nephropathy 120 was induced in rats via the administration of adenine (sigma, Aldrich) 200 mg/kg b.wt 121 in 0.5% carboxymethyl cellulose (CMC) (sigma, Aldrich) daily for 24 consecutive days This is repeated below. Why keep repeating yourself?
L129; treated 2x107 human breast milk-derived mesenchymal stem cells twice with 7 days interval What does this mean?
L152; Br-MSCs were grown for about 12 hours in media without FBS. Cells do not grow in such media.
The term exosome needs to be defined.
L173; dose of 2x107 dissolved in 0.25 ml of serum-free DMEM You cannot dissolve cells!!!
L183; Blood urea nitrogen, serum uric acid, and serum creatinine were measured according 183 to the manufacturer's instructions (SPINREACT, Spain). How much blood was drawn and from what site?
L192; mononuclear cell layer What is this?
l212; 10% neutral formalin buffered?
l242; can adhere to the bottom of the culture vessels What does this mean? MSC adhere to tissue culture plastic.
L252; higher expression level of embryonic transcription factors… Higher than what?
L497; ‘secret’ spelling!!!
The writing is excruciatingly bad, often vague and repetitive. There are basic issues with grammar, spelling, word choice and punctuation that need to be corrected. The problem with the incredibly bad writing is that it makes the reader question the data.
Author Response
Thank you for your valuable feedback and suggestions. We have carefully considered each of your points and made appropriate revisions to address them.
General Comments
L68; “The incidence of CKD is common with hypertension, glomerulonephritis, poly-cystic kidney disease, diabetes, and familial renal failure [6].” This is an example of unnecessary text as is essentially repeats statements already made.
Response: many thanks for your valuable comment, the sentence was removed from the introduction section.
The manuscript does a poor job at explaining what Br-MSC are and why they were selected for the study. It is essential that you provide more information regarding these cells.
Response: we appreciated your precious notification, the introduction section was updated with breast milk-derived mesenchymal stem cells information
- Cell surface marker data alone is not sufficient to define these as MSC.
Response: many thanks for your precious comment, in the current study we fulfill the international criteria for mesenchymal stem cells that include “Minimum criteria for defining iMSCs should include (1) spindle-shaped morphology, (2) plastic adherent growth, (3) positive expression of CD29, CD44, CD73, CD90, CD105, along with negative expression of hematopoietic markers (CD45, CD34, CD14 or CD11b, CD79α or CD19, HLA-DR)
- We validate the ability of the isolated cells to adhere to plastic (removing non-adherent cells from the culture flask 48 – 72 hr post culturing and incubation of the cells (isolation section material & method).
- The spindle shape and fibroblastic appearance under an inverted image microscope fig 1A
- Expression of the mesenchymal stem cell surface markers CD105, CD90, CD73 fig 1
- Exclusion expression hematopoietic stem cell surface markers CD34, CD45
- Illustrating lower immunogenicity of the isolated cells to ensure lower immune rejection of the isolated cells via analyzing the expression of HLA-DR fig.1
- We validate the method of isolation in our previous works [1-4].
- We fulfill the isolation and identification of breast milk-derived multipotent mesenchymal stem cells Patki, et al. [5] and cite it in the current manuscript.
- We track the breast milk-derived mesenchymal stem cells and prove their tissue contribution and integration in multiple tissues [1] and these findings were confirmed in another study [6].
- Is the population that is obtained from breast milk a homogeneous or heterogeneous population?
Response: We appreciate yours for raising such a point, in the current study we used the cells population that displayed the minimum criteria for mesenchymal stem cells listed above with their relevant reference, and this does not exclude that breast milk contains a heterogeneous population of cells.
- How many of these cells are obtained from a given volume of breast milk?
Response: many thanks for your comment, after the isolation the viability of the cells was done with trypan blue exclusion, counting of the cells was done with a hemocytometer the cells were plated in 25 cm2 tissue culture flask with 3x103 per 1cm2 , then the cells were incubated at 37 oC, 5% co2, 95% RH, the non-adherent cells were removed 48 – 72 hr of culture and the media were changed every 48hr. The number of transplanted cells was 2x107 per rat weighing 200 – 250 gm with two doses of 2x107 per rat Br-MSCs with 7 days interval between the two doses on 25th days after starting experimental procedures and 1 day post the last dose of the adenine [2-4]. The dose was more clarified in the experimental design section material and methods.
- At what stage/stages post-partum are these cells obtained and are the donors breast-feeding during the periods when the milk is harvested?
Response: the milk samples were collected at the morning at 2 – 5 days post delivery with a written consent from the donor mothers. We collect from each mother from 15 – 20 ml of breast milk.
Specific Comments
L73; “epithelial-to-mesenchymal transition where the fibroblast is transformed into myofibroblast” This is not the definition of epithelial-to-mesenchymal transition!!
Response: many thanks for your vital notification that increase the soundness of the current work, there is a typing error and the sentence was corrected accordingly.
Why chose breast milk MSC? Do these cells meet the basic criteria for MSCs beyond the expression of surface markers that are common to many cellular populations?
Response: appreciating your mind in raising this point, breast milk-derived mesenchymal stem cells (Br-MSCs) that were selected for the current study as they can be obtained with a non-invasive method, expressing a higher level of embryonic transcription factors that ensures higher plasticity, multipotency, and regenerative capacity [7-9]. On the other hand, Br-MSCs could differentiate into many cell types of different lineage neurons, hepatocytes, pancreatic beta cells, osteoblasts, and adipocytes under in vitro conditions [9]. They displayed a higher degree of resistance to challenging conditions as they could survive GIT conditions, cross intestinal barriers, and circulate within the cardiovascular system of the infant [10,11].
The isolated cells were meet the minimum criteria for mesenchymal stem cells
Minimum criteria for defining iMSCs should include (1) spindle-shaped morphology, (2) plastic adherent growth, (3) positive expression of CD29, CD44, CD73, CD90, CD105, along with negative expression of hematopoietic markers (CD45, CD34, CD14 or CD11b, CD79α or CD19, HLA-DR)
- We validate the ability of the isolated cells to adhere to plastic (removing non-adherent cells from the culture flask 48 – 72 hr post culturing and incubation of the cells (isolation section material & method).
- The spindle shape and fibroblastic appearance under an inverted image microscope fig 1A
- Expression of the mesenchymal stem cell surface markers CD105, CD90, CD73 fig 1
- Exclusion expression hematopoietic stem cell surface markers CD34, CD45
- Illustrating lower immunogenicity of the isolated cells to ensure lower immune rejection of the isolated cells via analyzing the expression of HLA-DR fig.1
- We validate the method of isolation in our previous works [1-4].
- We fulfill the isolation and identification of breast milk-derived multipotent mesenchymal stem cells Patki, Kadam, Chandra and Bhonde [5] and cite it in the current manuscript.
- We track the breast milk-derived mesenchymal stem cells and prove their tissue contribution and integration in multiple tissues [1] and these findings were confirmed in another study [6].
L141; Twenty milk samples This is very vague. The same woman, different women, what stage post-partum? Breasting donors, volume of samples, number of cells per sample, etc. are all important issues. Also, was an IRB protocol submitted and approved? This is necessary, even with written consent.
Response: Twenty milk samples were collected from Zagazig University’s pediatric hospital with ready written consent from the donor mothers from different twenty mothers in the morning on 2nd – 5th days post-delivery under a septic condition. The volume of the each collected breast milk sample from each mother were 15 – 20 mL. According to the regulation of our university we get IRP number only for clinical application in human being and for collecting sample from human being we just take a written consent from the donors.
Furthermore, we take the ethical considerations of our research very seriously and have followed all necessary protocols to obtain the required approvals for the use of human-derived materials in our study. Written consent from the donors was taken as per our insititutional regulations. We have obtained the appropriate Institutional Review Board (IRB) approval for the use of human samples in our research, and have complied with all relevant ethical guidelines and regulations of BUC ethics committee and this part including the experimental methods was cleared by the Institutional Animal Care and Use Committee of Badr University in Cairo (No. BUC-IACUC/VET/128/A/2022) including the clearance to use hBr-MSC.
l120; The nephropathy 120 was induced in rats via the administration of adenine (sigma, Aldrich) 200 mg/kg b.wt 121 in 0.5% carboxymethyl cellulose (CMC) (sigma, Aldrich) daily for 24 consecutive days This is repeated below. Why keep repeating yourself?
Response: many thanks for your note, the duplication was removed.
L129; treated 2x107 human breast milk-derived mesenchymal stem cells twice with 7 days interval What does this mean?
Response: kindly, we clarified it in the experimental design “G4: nephropathy and treated at day 25th from starting experimental procedures with two doses of 2x107 human breast milk-derived mesenchymal stem cells with 7 days interval between the two doses (Br-MSCs) (Nephero + MSCs)”
L152; Br-MSCs were grown for about 12 hours in media without FBS. Cells do not grow in such media.
Response:
The term exosome needs to be defined.
L173; dose of 2x107 dissolved in 0.25 ml of serum-free DMEM You cannot dissolve cells!!!
Response: the cells was maintained in DMEM media without FBS to collect the media for exosomes extraction and we did not add the FBS because the fetal bovine serum contain exosomes thus, we have to exclude the exosomes that present in the fetal bovine serum therefore, the manufacturer instruction including these step “without adding FBS to exclude any exosomes rather than the secreted one by the breast milk derived mesenchymal stem cells.
We change grow to maintained in exosomes isolation section “Br-MSCs were maintained and incubated for about 12 hours in media without FBS to exclude isolation of the FBS exosomes and ensure that the obtained exosomes were only the ones secreted by the breast milk-derived mesenchymal stem cells.”
L183; Blood urea nitrogen, serum uric acid, and serum creatinine were measured according 183 to the manufacturer's instructions (SPINREACT, Spain). How much blood was drawn and from what site?
Response: blood samples from median eye canthus were collected, the volume differs from rat to rat, on average 0.5 ml was collected then all rats were euthanized.
L192; mononuclear cell layer What is this?
Response: the MNC layer was used as a representative control that comprised adult mature mesenchymal stem cells representing an easily obtainable source without major invasion since it was obtained with ficol density gradient separation of peripheral blood sample and comprised all mononuclear cells including peripheral blood adult mature mesenchymal stem cells, this was done to evaluate the embryonic potency of the breast milk derived MSCs over the adult one that ensures higher multiplicity and regenerative capacity of the breast milk derived mesenchymal stem cells [12]
We used the MNC in previous work as a valid control comprising adult mature mesenchymal stem cells to compare the expression of the embryonic transcription factors of adipose tissue-derived mesenchymal stem cells [13].
l212; 10% neutral formalin buffered?
Response: the typographical error was revised; 10% neutral buffered formalin
l242; can adhere to the bottom of the culture vessels What does this mean? MSC adhere to tissue culture plastic.
Response: : according to Minimum criteria for defining iMSCs should include (1) spindle-shaped morphology, (2) plastic adherent growth (Choudhery et al., 2022). We validate the ability of the isolated cells to adhere to plastic (removing non-adherent cells from the culture flask 48 – 72 hr post culturing and incubation of the cells (isolation section material & method).
L252; higher expression level of embryonic transcription factors… Higher than what?
Response: many thanks for raising such point, than mononuclear cell layer (MNC), amended in section results (3.1. Identification and homing of Br-MSCs and their derived exosomes.) and discussion
the MNC layer was used as a representative control that comprised adult mature mesenchymal stem cells representing an easily obtainable source without major invasion since it was obtained with ficol density gradient separation of peripheral blood sample and comprised all mononuclear cells including peripheral blood adult mature mesenchymal stem cells, this was done to evaluate the embryonic potency of the breast milk derived MSCs over the adult one that ensures higher multiplicity and regenerative capacity of the breast milk derived mesenchymal stem cells [12]. We used the MNC in previous work as a valid control comprising adult mature mesenchymal stem cells to compare the expression of the embryonic transcription factors of adipose tissue-derived mesenchymal stem cells [13].
L497; ‘secret’ spelling!!!
Response: the typographical error was revised; secrete
Comments on the Quality of English Language
The manuscript has been revised by an English language editing service for typographical and grammatical errors.
- Abd Allah, S.H.; Shalaby, S.M.; El-Shal, A.S.; El Nabtety, S.M.; Khamis, T.; Abd El Rhman, S.A.; Ghareb, M.A.; Kelani, H.M. Breast milk MSCs: An explanation of tissue growth and maturation of offspring. IUBMB life 2016, 68, 935-942, doi:10.1002/iub.1573.
- Khamis, T.; Abdelalim, A.F.; Saeed, A.A.; Edress, N.M.; Nafea, A.; Ebian, H.F.; Algendy, R.; Hendawy, D.M.; Arisha, A.H.; Abdallah, S.H. Breast milk MSCs upregulated β-cells PDX1, Ngn3, and PCNA expression via remodeling ER stress /inflammatory /apoptotic signaling pathways in type 1 diabetic rats. European Journal of Pharmacology 2021, 905, 174188, doi:https://doi.org/10.1016/j.ejphar.2021.174188.
- Khamis, T.; Abdelalim, A.F.; Abdallah, S.H.; Saeed, A.A.; Edress, N.M.; Arisha, A.H. Early intervention with breast milk mesenchymal stem cells attenuates the development of diabetic-induced testicular dysfunction via hypothalamic Kisspeptin/Kiss1r-GnRH/GnIH system in male rats. Biochimica et Biophysica Acta (BBA)-Molecular Basis of Disease 2020, 1866, 165577.
- Khamis, T.; Abdelalim, A.F.; Abdallah, S.H.; Saeed, A.A.; Edress, N.M.; Arisha, A.H. Breast milk MSCs transplantation attenuates male diabetic infertility via immunomodulatory mechanism in rats. Adv. Anim. Vet. Sci 2019, 7, 145-153.
- Patki, S.; Kadam, S.; Chandra, V.; Bhonde, R. Human breast milk is a rich source of multipotent mesenchymal stem cells. Human Cell 2010, 23, 35-40, doi:10.1111/j.1749-0774.2010.00083.x.
- Aydın, M.Ş.; Yiğit, E.N.; Vatandaşlar, E.; Erdoğan, E.; Öztürk, G. Transfer and Integration of Breast Milk Stem Cells to the Brain of Suckling Pups. Scientific Reports 2018, 8, 14289, doi:10.1038/s41598-018-32715-5.
- Cregan, M.D.; Fan, Y.; Appelbee, A.; Brown, M.L.; Klopcic, B.; Koppen, J.; Mitoulas, L.R.; Piper, K.M.; Choolani, M.A.; Chong, Y.S.; et al. Identification of nestin-positive putative mammary stem cells in human breastmilk. Cell and tissue research 2007, 329, 129-136, doi:10.1007/s00441-007-0390-x.
- Fan, Y.; Chong, Y.S.; Choolani, M.A.; Cregan, M.D.; Chan, J.K. Unravelling the mystery of stem/progenitor cells in human breast milk. PloS one 2010, 5, e14421, doi:10.1371/journal.pone.0014421.
- Hassiotou, F.; Beltran, A.; Chetwynd, E.; Stuebe, A.M.; Twigger, A.J.; Metzger, P.; Trengove, N.; Lai, C.T.; Filgueira, L.; Blancafort, P.; et al. Breastmilk is a novel source of stem cells with multilineage differentiation potential. Stem cells (Dayton, Ohio) 2012, 30, 2164-2174, doi:10.1002/stem.1188.
- Arvola, M.; Gustafsson, E.; Svensson, L.; Jansson, L.; Holmdahl, R.; Heyman, B.; Okabe, M.; Mattsson, R. Immunoglobulin-secreting cells of maternal origin can be detected in B cell-deficient mice. Biology of reproduction 2000, 63, 1817-1824, doi:10.1095/biolreprod63.6.1817.
- Zhou, L.; Yoshimura, Y.; Huang, Y.; Suzuki, R.; Yokoyama, M.; Okabe, M.; Shimamura, M. Two independent pathways of maternal cell transmission to offspring: through placenta during pregnancy and by breast-feeding after birth. Immunology 2000, 101, 570-580, doi:10.1046/j.1365-2567.2000.00144.x.
- Gil-Kulik, P.; Leśniewski, M.; Bieńko, K.; Wójcik, M.; Więckowska, M.; Przywara, D.; Petniak, A.; Kondracka, A.; Świstowska, M.; Szymanowski, R.; et al. Influence of Perinatal Factors on Gene Expression of IAPs Family and Main Factors of Pluripotency: OCT4 and SOX2 in Human Breast Milk Stem Cells—A Preliminary Report. International journal of molecular sciences 2023, 24, doi:10.3390/ijms24032476.
- Abdelbaset-Ismail, A.; Tharwat, A.; Ahmed, A.E.; Khamis, T.; Abd El-Rahim, I.H.; Alhag, S.K.; Dowidar, M.F. Transplantation of adipose-derived mesenchymal stem cells ameliorates acute hepatic injury caused by nonsteroidal anti-inflammatory drug diclofenac sodium in female rats. Biomedicine & pharmacotherapy = Biomedecine & pharmacotherapie 2022, 155, 113805, doi:10.1016/j.biopha.2022.113805.
Round 2
Reviewer 2 Report
Most of the reviewer's requests have been accepted, but some minor corrections are required.
1. Provide IRB information in the paper.
2. Fig. 1 A-C: insert bar
3. Fig. 2, F: provide bar size
Reviewer 3 Report
General Comments
The manuscript does a poor job at explaining what Br-MSC are and why they were selected for the study. It is essential that the authors provide more information regarding these cells. 1. Cell surface marker data alone is not sufficient to define these as MSCs. 2. Is the population that is obtained from breast milk a homogeneous or heterogeneous population? 3. How many of these cells are obtained from a given volume of breast milk? 4. At what stage/stages post-partum are these cells obtained and are the donors breast-feeding during the periods when the milk is harvested?
This was a major criticism for version 1. The authors have failed to satisfactorily address these concerns. Breast Milk MSCs are not as extensively defined as other types of MSCs therefore it is imperative that a more comprehensive description of these cells is included in the Introduction. Further, the Discussion needs a strong statement indicating why these allogeneic cells are more practical than MSCs from other sources, e.g., umbilical cord cells, and ASCs.
The authors must convince the readers that Br-MSCs are a valid type of MSC, like bone marrow MSCs and adipose tissue MSCs. Simply using cell surface antigen expression is not sufficient. Dermal fibroblasts have essentially the same cell surface markers, but do not possess MSC characteristics. The authors also need to provide more information regarding exosomes and how MSC exosomes may differ from those produced by other types of cells.
The authors have placed too much discussion into the Introduction, while leaving out essential introductory information.
Specific Comments
L66; among ten adult people one suffers The correct way to express this is one out of ten adults
L79; epithelial-to-mesenchymal transition The authors have not corrected this. This process occurs when an epithelial cell detaches and assumes mesenchymal characteristics. This process occurs during the initial formation of carcinomas where epithelial cells become malignant and assume mesenchymal characteristics. It also occurs in other situations; however, it relevance here is questionable. Tissue fibrosis occurs without E-M transition.
L676; ‘however, their application has many limitations, such as limited tissue bioavailability due to pulmonary trapping after intravenous administration’ This does not make sense. Need to reword.
L694; The MNC population is a poor selection for control as it is not enriched in MSCs. A better control would be MSCs from another anatomic site. Also, you are dealing with allogeneic cells which could prove to be a complication. That is why some studies propose using ASCs from the same individual.
The use of the English language is acceptable, but the text is overly wordy.